# HSP90-CDC37-PP5 forms a structural platform for kinase dephosphorylation

Jasmeen Oberoi [1], Xavi Aran Guiu [1], Emily A. Outwin[1], Pascale Schellenberger[2], Theodoros I. Roumeliotis [3], Jyoti S. Choudhary [3] & Laurence H. Pearl [1,3] ✉

Activation of client protein kinases by the HSP90 molecular chaperone system is affected by phosphorylation at multiple sites on HSP90, the kinase-specific co-chaperone CDC37, and the kinase client itself. Removal of regulatory phosphorylation from client kinases and their release from the HSP90-CDC37 system depends on the Ser/Thr phosphatase PP5, which associates with HSP90 via its N-terminal TPR domain. Here, we present the cryoEM structure of the oncogenic protein kinase client BRAF$^{V600E}$ bound to HSP90-CDC37, showing how the V600E mutation favours BRAF association with HSP90-CDC37. Structures of HSP90-CDC37-BRAF$^{V600E}$ complexes with PP5 in autoinhibited and activated conformations, together with proteomic analysis of its phosphatase activity on BRAF$^{V600E}$ and CRAF, reveal how PP5 is activated by recruitment to HSP90 complexes. PP5 comprehensively dephosphorylates client proteins, removing interaction sites for regulatory partners such as 14-3-3 proteins and thus performing a 'factory reset' of the kinase prior to release.

Interaction with the HSP90 molecular chaperone system is a prerequisite for the stability and biological function of a large proportion of the kinome[1], including most of the main oncogenic protein kinases[2]. Recruitment of kinases to the HSP90 system is mediated by CDC37[3], which functions as an adaptor able to interact independently with HSP90 and protein kinases and facilitate their association[4]. CDC37 is subject to a number of phosphorylation events[5], one of which—phosphorylation of Ser13 by casein kinase 2 (CK2)[6,7]—is critical to its function in protein kinase activation. HSP90 itself is also multiplied phosphorylated[8], and while none are critical to its core biochemistry, several of the modified sites have nonetheless been shown to have important functions in the regulation of ATP-utilisation and/or co-chaperone and client interactions[9–13]. The protein kinase clients of HSP90–CDC37 are themselves frequently phosphorylated, sometime autogenously, as part of their regulation, and can, in turn, participate in the phosphorylation of components of the chaperone complexes to which they are recruited, generating a complex network structure of post-translational regulation—the so-called Chaperone Code—the surface of which has only been scratched[14].

Phosphorylation is, by its nature, a reversible post-translational modification, and its role in switching the behaviour of a modified protein depends both on the kinase that 'writes' the modification and the phosphatase that 'erases' it[15]. HSP90 is directly associated with an unusual serine/threonine protein phosphatase PP5 (Ppt1p in yeast), which has been implicated in the maturation/activation of a number of HSP90-dependent client proteins[16–19]. PP5 has a tetratricopeptide (TPR) domain attached to the N-terminus of an Mn$^{2+}$-dependent PP1/PP2A/PP2B family phosphatase domain[20]. In common with several other HSP90-associated proteins, the TPR domain confers a high affinity for the MEEVD motif that forms the extreme C-terminus of HSP90[21]. Ppt1p, the yeast homologue of PP5, has been implicated in regulating the phosphorylation of HSP90 itself, with a deficit of Ppt1p activity leading to reduced activation of a range of client proteins in vivo[22]. Within the specific context of the HSP90–CDC37 system, activation of protein kinase clients in vivo has been shown to depend on dephosphorylation of pSer-13 in CDC37 by PP5, which only occurs when PP5 and CDC37 are bound simultaneously to the same HSP90 dimer[23].

[1]Genome Damage and Stability Centre, School of Life Sciences, University of Sussex, Falmer, Brighton BN1 9RQ, UK. [2]Electron Microscopy Imaging centre, School of Life Sciences, University of Sussex, Falmer BN1 9QG, UK. [3]Institute of Cancer Research, Chester Beatty Laboratories, 237 Fulham Road, London SW3 6JB, UK. ✉e-mail: laurence.pearl@sussex.ac.uk

To understand how PP5 operates in the context of an HSP90−CDC37 client complex, we have reconstituted an active PP5 complex with HSP90, CDC37 and the highly HSP90-dependent oncogenic V600E mutant of the protein kinase BRAF[24,25]. We have determined the cryoEM structure of an HSP90−CDC37−BRAF[V600E] complex, and structures of HSP90−CDC37−BRAF[V600E] with PP5 bound in phosphatase-engaged and phosphatase-autoinhibited conformations. These structures reveal how single PP5 docks with the dimeric C-terminus of HSP90, and how docked PP5 rearranges to allow the catalytic phosphatase domain to access phosphorylation sites on the chaperone, co-chaperone and client. Together with proteomic analysis of PP5 activity on HSP90−CDC37-bound BRAF[V600E] and HSP90−CDC37-bound CRAF, our studies reveal how the HSP90−CDC37−PP5 complex acts to comprehensively dephosphorylate the bound client.

## Results

### PP5 dephosphorylates CDC37 within an HSP90−CDC37−BRAF[V600E] complex

We previously showed that protein phosphatase 5 (PP5) was able to dephosphorylate pSer13 in CDC37 when both proteins were physically associated with HSP90[23]. To determine whether PP5 could also do this when CDC37 was engaged in a complex with a client protein and HSP90, we co-expressed and purified an HSP90−CDC37−BRAF[V600E] (HCK) complex from insect cells (see Methods), and used a phosphospecific antibody to demonstrate that wild-type PP5 could dephosphorylate CDC37−pSer13 in the complex in a time-dependent manner (Supplementary Fig. 1A).

To attempt to trap a productive complex of PP5 engaged with HSP90−CDC37−BRAF[V600E], we incubated the HSP90−CDC37−BRAF[V600E] complex with a PP5 D274N mutant which had previously been shown to catalytically inactivate PP5 with minimal disruption to substrate binding[26] and were able to purify a stable HSP90−CDC37−BRAF[V600E]−PP5 complex (HCK-P) on size exclusion chromatography (Supplementary Fig. 1B).

### CryoEM structure determination

For structural studies, the purified complex was cross-linked (Supplementary Fig. 1C) and applied to cryoEM grids, which were then plunged into a liquid ethane/propane mixture. Movies from selected regions of the grids were recorded on an FEI Titan Krios microscope equipped with a Falcon IV detector (see Methods). Movies were motion corrected, images processed, and particles picked using cryoSPARC[27] and RELION 4.0[28] (Supplementary Fig. 1D, E). We obtained particle sets representative of three different structures, which were separately refined. Final maps for HCK, HCK-P$_{open}$, and HCK-P$_{closed}$ complexes had overall resolutions of 3.4, 4.2, and 3.9 Å, respectively and allowed the fitting of substantive atomic models using the known crystal and cryoEM structures of the components (see Methods). The image processing workflow is shown in Supplementary Fig. 2.

### Structure of HSP90−CDC37−BRAF[V600E]

The structure of the HSP90−CDC37−BRAF[V600E] complex consists of two molecules of human HSP90β arranged in the ATP-bound closed conformation originally observed in a complex of yeast HSP90 and the co-chaperone P23/Sba1[29] (Fig. 1A). The polypeptide chain for both HSP90β molecules can be traced through more or less continuous, ordered density from Glu10 to Glu692, with the exception of the low-complexity 'linker segment' from approximately 220–275 which connects the N-terminal and central domains. Consistent with the closed conformation, bound ATP (or ADP-molybdate) is present in the N-terminal domains of both HSP90 molecules (Fig. 1B).

CDC37 in the complex presents in a very similar conformation as seen in the cryoEM structure of the HSP90−CDC37−CDK4 complex[30], with the N-terminus (residues 1–120), which consists predominantly of

a long coiled-coil α-hairpin protruding from one side of the core HSP90 dimer, while the globular helical domain that forms the bulk of the C-terminal part (136–378)[31] is packed against the opposite face of the dimer (Fig. 1A). The two-halves of CDC37 are connected by an extended β-strand (121–135) which hydrogen bonds onto the edge of the central β-sheet of the middle domain of one of the HSP90 monomers. The polypeptide chain in the N-terminus can be traced from the N-terminal methionine to Cys54 and from Leu91–Glu134; however, the tip of the coiled-coil α-hairpin (residues 55–90) is not visible in the map. The C-terminal part of CDC37 is far less well defined than the N-terminus, with the structure only discernible at the level of secondary structural elements from residue 140 to residue 266, suggesting a high degree of disorder and/or multiple conformational states for this loosely bound domain.

Serine 13, whose phosphorylation and targeted dephosphorylation are critical for client kinase activation by HSP90[6,7,23], is clearly phosphorylated within the complex and engaged with the side chains of CDC37 residues His33 and Arg36, and Lys406 of HSP90 as previously seen in the cryoEM structure of the HSP90−CDC37−CDK4 complex[30] (Fig. 1C).

Although the complex was formed by co-expression of the full-length proteins, relatively little of the 84 kDa BRAF[V600E] is visible in the cryoEM volume, with only the C-terminal lobe of the kinase domain being well defined in the map (Fig. 1D). The polypeptide chain for this segment can be traced into clear features from Thr521 to Ile724, apart from the region corresponding to the 'activation segment'[32] connecting the 594-DFG-596 and 621-APE-623 motifs, which is poorly ordered. The final 42 residues at the C-terminus beyond the kinase C-lobe are also disordered.

The face of the BRAF[V600E] kinase C-lobe that forms one wall of the ATP-binding cleft in the fully folded kinase structure[33] interacts with a contiguous segment of CDC37 from Thr19 to Ala35, incorporating the beginning of the first helix in the coiled-coil segment (Fig. 1E). The core of the interface is provided by the side chains of His20, Ile23, Asp24, Ser27 and Trp31 of CDC37, which sit together in a channel in BRAF[V600E] lined by Arg562, Gly563, Tyr566, Leu567, Ile572, His574, Thr590, Lys591, Ile592, Gly593 and Asp594.

One consequence of the interaction of CDC37−Trp31 with BRAF[V600E] is to force the catalytically important DFG motif, into a quite different conformation to that found in the folded active kinase, with the following activation segment containing the oncogenic V600E mutated residue, being disordered. V600E and other common oncogenic Val600 mutations have been shown to confer a strong dependence on association with HSP90−CDC37 for cellular stability and activation, whereas wild-type BRAF is a relatively weak 'client'[24]. Val600 in wild-type BRAF forms part of a hydrophobic cluster that holds the activation segment in an ordered inhibitory conformation[34], which is destabilised by oncogenic mutations such as V600E[33,35], contributing to unregulated kinase activity. Such destabilisation would also facilitate the conformational switch of the DFG and attached activation segment required by the interaction with CDC37 seen here, more readily than the hydrophobic and more rigid wild-type sequence, providing a satisfactory explanation for the substantially stronger HSP90-dependence of the oncogenic BRAF mutants.

HSP90 makes only a few direct contacts with the BRAF[V600E] kinase C-lobe, restricted to peripheral interactions with surface-exposed side chains of Arg338, Phe341 and Trp312 from the central region of HSP90 (Fig. 1F)−the latter two previously implicated in client interactions in an earlier mutagenesis study[36], and a polar interaction between HSP90−Arg196 and BRAF−Asp565. The major interactions between HSP90 and the kinase client involve residues 521–533 of BRAF[V600E], which would be part of the N-terminal lobe in the fully folded kinase structure, which in the complex threads between the central segment of the two HSP90 monomers

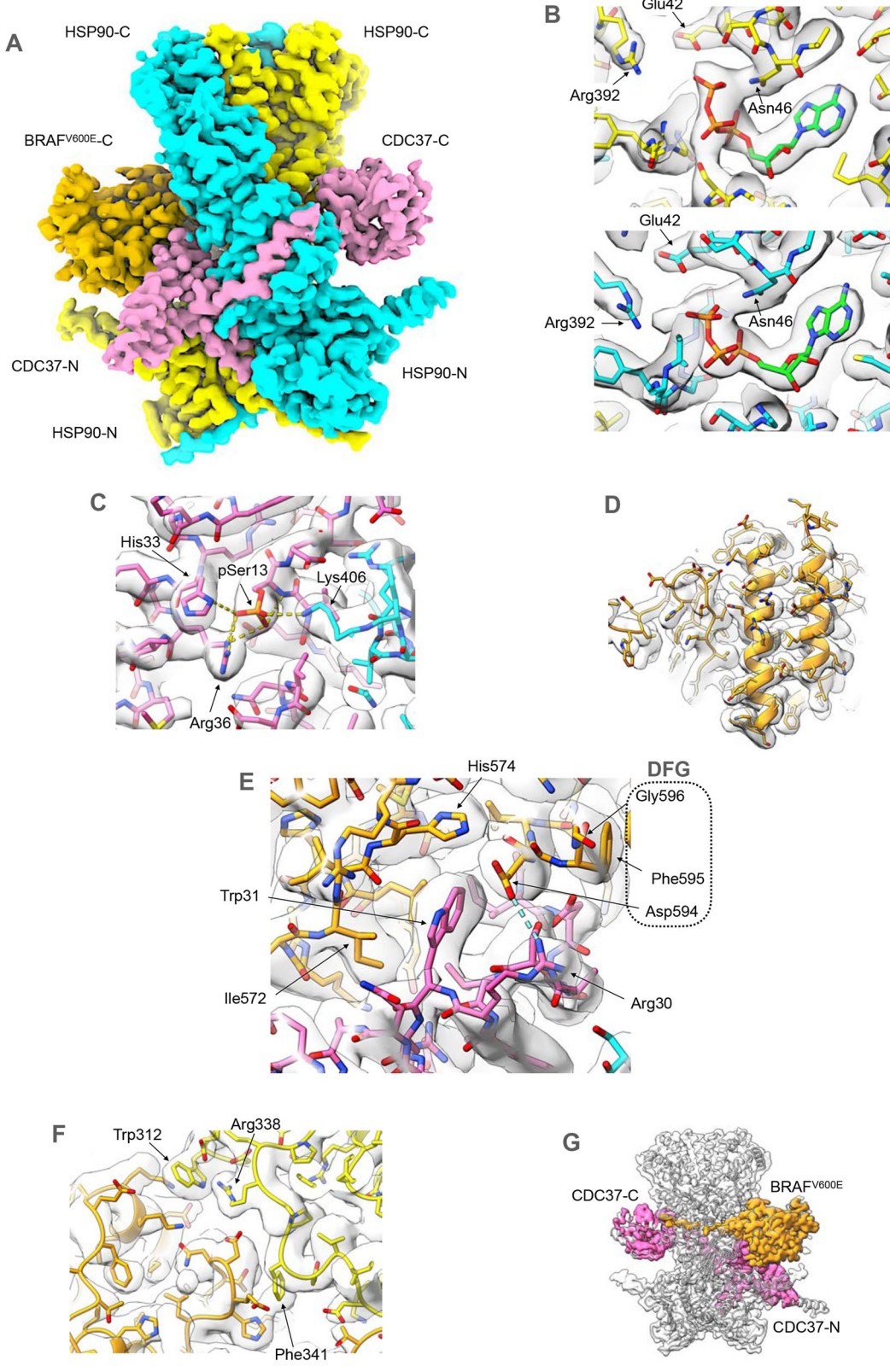

adjacent to the extended loops from Asn351 to Phe344 that come close together at the heart of the HSP90 dimer (Fig. 1G). Upstream of BRAF[V600E] residue 521, the chain emerges on the opposite face of the dimer, to run adjacent to the loosely bound globular domain in the C-terminal half of CDC37. However, the map in this region lacks detail due to conformational flexibility, and/or the presence of multiple conformations.

## Structure of HSP90–CDC37–BRAF[V600E]–PP5 complexes

Two different sets of particles were obtained in which additional volume corresponding to PP5 was evident bound to the C-terminus of the HSP90 dimer within the HSP90–CDC37–BRAF[V600E] complex. The two structures are distinguished by whether the C-terminal phosphatase domain of PP5 is packed against the N-terminal TPR domain in a 'closed' conformation or is substantially separated from it in an 'open'

**Fig. 1 | Structure of HSP90–CDC37–BRAF^V600E complex. A** Experimental cryoEM map of HSP90–CDC37–BRAF^V600E complex, surface coloured according to the underlying protein chain. The HSP90 dimer (blue and yellow—the colours of the Ukrainian flag) is in a closed conformation, with CDC37 (pink) wrapping around the edge of one of the HSP90 monomers. The C-lobe of the kinase domain of BRAF^V600E (orange) packs against the opposite face to the globular C-terminal half of CDC37, and interacts with the N-terminal coiled-coil α-hairpin of CDC37. This and all other molecular graphics were created using ChimeraX[61]. **B** ATP (or possibly ADP-molybdate) molecules (green bonds) are bound in the N-terminal domain of both HSP90 monomers and interact with the catalytic Arg392 from the middle domain. **C** Phosphorylated Ser13 of CDC37 interacts with His33 and Arg36 of CDC37, stabilising the conformation of the N-terminal part of CDC37 and bridging to Lys406 from the middle segment of one of the HSP90 monomers. **D** The bound BRAF^V600E is well resolved throughout, allowing for a nearly full tracing of its amino acid sequence in the cryoEM map. **E** Within the complex, Trp31 displaces Phe595 of the key regulatory DFG motif into a different conformation than in the intact kinase, stabilised by the interaction of DFG Asp594 with CDC37 Arg30. This conformational switch is facilitated by the oncogenic V600E mutation in the 'activation segment' immediately following the DFG motif, and explains why the oncogenic BRAF^V600E mutant is a strong client of the HSP90-CDC37 chaperone system, whereas the wild-type is not[24]. **F** HSP90 itself only makes peripheral contact with the kinase C-lobe, but mutation of the HSP90 residues involved impairs kinase activation in vivo[36]. **G** As previously seen for CDK4 in complex with HSP90 and CDC37[30], the strand from the kinase N-lobe immediately upstream of the well-ordered C-lobe, becomes linearised and stretches between the two HSP90 monomers to emerge on the other face of the complex adjacent to the globular part of CDC37. No ordered structure upstream of this is visible in the cryoEM maps.

conformation. The locations and conformation of the HSP90 monomers and the visible parts of CDC37 and BRAF^V600E are essentially the same as in the HSP90–CDC37–BRAF^V600E complex (see above).

In the PP5-closed conformation (Fig. 2A), the convex face of the TPR domain is juxtaposed with the active site of the phosphatase domain in a similar manner to that previously observed in the crystal structure of an autoinhibited conformation of PP5[37] (Fig. 2B). The concave face of the TPR domain is directed away from the phosphatase domain, and comparison with the ensemble NMR structure of a complex of PP5-TPR with the C-terminal -MEEVD peptide of HSP90[21] suggests that the channel formed by the TPR helices on this side of the domain is occupied by a bound C-terminal -MEEVD peptide of HSP90 in the cryoEM structure, although this cannot be resolved in detail (Fig. 2C). The flexible segment linking the C-terminal peptide to the globular core of HSP90 is not visible. An additional interface with HSP90 is made by the distal end of the elongated last α-helix of the PP5-TPR domain[38], centred on PP5-Phe148, which packs into a hydrophobic pocket at the C-terminus of the HSP90 dimer close to the dimer interface lined by Ile684, Gly687 and Leu688 from the HSP90 monomer most likely providing the interacting MEEVD peptide, and Ala650, Asp653 and Leu654 of the other HSP90 monomer (Fig. 2D).

In the PP5-open conformation (Fig. 3A), the two domains of PP5 are completely separated, with the C-terminus of the TPR domain at Arg150 more than 50 Å away from the N-terminus of the globular phosphatase domain at Tyr176. As with the closed complex, the cryoEM map suggests the presence of the HSP90 C-terminal MEEVD peptide bound into the concave face of the TPR, and the terminal helix of the TPR domain makes the equivalent interaction with the hydrophobic pocket formed by the C-terminal domains of the HSP90 dimer (Fig. 3B).

The detached phosphatase domain binds down towards the middle of the complex, bridging between surface loops at 461–467 in the middle domain of one HSP90 monomer and 569–574 in the C-terminal domain of the other monomer. In this position, the substrate-binding cleft of the phosphatase domain docked into the cryoEM volume, would be in direct contact with the HSP90-bound C-lobe of the BRAF^V600E close to several inter-helix loops, and the point at which the unstructured C-terminus of BRAF^V600E would extend from the globular C-lobe. Although the resolution of the map in this region does not allow full modelling beyond the globular C-lobe, the early part of this BRAF^V600E segment downstream of residue 721, including pSer729, would be well positioned to interact productively with the phosphatase active site, although much higher resolution will be required to confirm this (Fig. 3C).

As HSP90 is dimeric, there are two symmetrical disposed copies of the C-terminal hydrophobic binding site that the PP5 TPR domain interacts with. The binding of CDC37 and the kinase client render the overall complex asymmetrical, but as these are bound by the middle domain, the twofold symmetry of the HSP90 C-terminus is largely unaffected. However, the two sites are sufficiently close together that

the binding of PP5 to one site sterically occludes the other, thereby restricting the stoichiometry to a single PP5 per HSP90 complex.

Fascinatingly, the PP5 TPR domain in the closed complex binds to one site such that the phosphatase domain is held on the face of the HSP90 dimer that presents the globular domain of CDC37, while in the open complex, the TPR binds to the symmetry equivalent site so that the phosphatase domain is on the face that presents the kinase C-lobe and the coiled-coil helical hairpin of CDC37.

## PP5 phosphatase targets
While dephosphorylation of CDC37-pSer13 is the best studied HSP90-associated activity of PP5[23] (see Supplementary Fig. 1A), under the conditions in which HSP90–CDC37–BRAF^V600E is expressed and purified to be amenable to structural studies, CDC37–pSer13 is fully buried in the core of the ATP-bound closed HSP90 complex and remains so in the presence of the catalytically dead PP5. Even though the phosphatase domain of PP5 can detach from the C-terminus of the HSP90 dimer and move substantially towards CDC37, pSer13 would only become accessible when the N-terminal domains of HSP90 separate following ATP hydrolysis, so trapping a structurally tractable complex in which PP5 is engaged with CDC37–pSer13 remains to be achieved.

However, CDC37 is not the only component of the complex that is susceptible to phosphorylation, and, therefore, a potential substrate for PP5. To gain some insight into other potential substrates, we mapped the phosphorylation sites on the purified HSP90–CDC37–BRAF^V600E complex with and without treatment with PP5, by mass spectrometry (see Methods, Fig. 4A, C). We identified two sites in HSP90 (Ser226, Ser255), which were significantly diminished by PP5 treatment. Both of these CK2 sites are within the charged linker segment connecting the N and middle domains of HSP90 and have been implicated in the regulation of HSP90β secretion[39]. We identified multiple sites in BRAF^V600E whose phosphorylation was significantly ($p < 0.05$) decreased by PP5 treatment (Fig. 4A). One (Ser151) occurs just before the RAS binding domain (RBD), while six (Ser339, Ser365, Thr401, Ser429, Ser432, Ser446) occur within the disordered segment between the RBD and kinase domains. Ser365 plays a critical role in 14-3-3 binding[34] and, along with Ser429, has been shown to have differential regulatory effects on different BRAF isoforms[40], while Ser446, which maps just upstream of the kinase N-lobe, is the topological equivalent of Ser338 in CRAF whose dephosphorylation by PP5 was previously shown to deactivate kinase signalling activity[16].

Within the kinase domain itself, which is the focus of interaction of the HSP90–CDC37 system, we identified no phosphorylation sites in the N-lobe, but two sites within the C-lobe, which is the only part of BRAF^V600E in complex with HSP90–CDC37 that is resolved in the cryoEM structure. Ser614, identified as inhibitory phosphorylation specifically enriched in the V600E mutant[41], maps to the C-terminal end of the activation segment 594–623, which is disordered in the structure, while Ser675 is involved in the regulation of BRAF ubiquitylation by the E3 ligase ITCH[42] is in the middle of an extended coil that

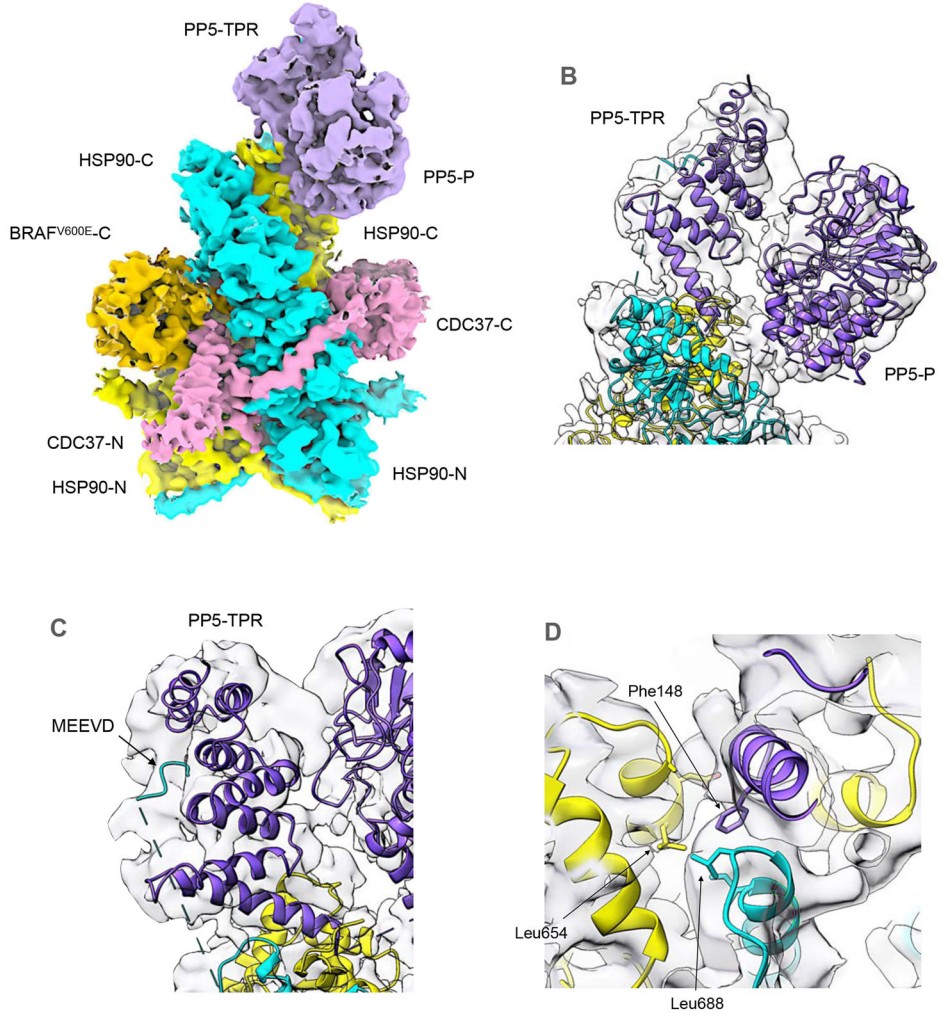

**Fig. 2 | Structure of HSP90–CDC37–BRAF^V600E complex with autoinhibited PP5.** Experimental cryoEM map of HSP90–CDC37–BRAF^V600E–PP5 complex, surface coloured as Fig. 1, but with the addition of PP5 (purple), bound to the C-terminal end of the HSP90 dimer. The resolution of the region of the map corresponding to PP5 does not allow ab initio tracing of the polypeptide chain, which has therefore been modelled by rigid-body docking of previously determined crystal structures for the globular regions into the corresponding volume. The fit of the molecular model was then optimised by real-space refinement. **A** PP5 is bound in the auto-inhibited closed conformation[37] in which the convex face of the TPR domain (PP5-TPR) occludes the substrate binding cleft of the phosphatase domain (PP5-P). As in

previous PP5 crystal structures, the flexible linker connecting the last elongated α-helix of the TPR domain and the start of the phosphatase domain is disordered. **B** Although the resolution is insufficient for direct molecular modelling, super-imposition of NMR structures of the isolated PP5-TPR domain in complex with HSP90 C-terminal peptides[21] on the cryoEM volume, suggests the presence of the C-terminal MEEVD sequence bound in the convex face of the TPR domain. The conformation of this peptide, as shown, remains speculative. **C** Additional to the HSP90–MEEVD interaction, the last helix of the PP5-TPR domain packs against a cluster of helices from the C-terminal domains of both HSP90 monomers, with the side chain of Phe148 at the tip of the last PP5 helix in a hydrophobic pocket.

connects two helices. Both of these residues map to parts of the surface of BRAF^V600E that are not involved in interaction with HSP90 or CDC37, and would therefore be accessible to the PP5 phosphatase domain given the flexibility of its connection to its TPR anchor. We found no phosphorylation of the activation segment residues Thr599 and Ser602, whose phosphorylation by MLK3[43] is required for full BRAF (wt or V600E) kinase activity[44].

Three further sites (Ser729, Ser750 and Thr753) all occur in the unstructured segment that follows the end of the C-lobe at residue 720. Ser729 has recently been shown to have a key role in 14-3-3 binding in concert with Ser365[34,45]. The proximity of the C-terminal end of the C-lobe to the substrate binding cleft of the phosphatase domain strongly suggests that one or more of these sites are engaged with the catalytically inactivated PP5.

We performed the same analysis for an HSP90–CDC37–CRAF complex, and as with the BRAF complex, observed a set of phosphorylation sites present on the kinase in the absence of PP5 treatment, that were significantly dephosphorylated when PP5 was added

(Fig. 4B, C). These included Ser233, Ser259 and Ser621, which, similarly to Ser365 and Ser729 in BRAF, mediate CRAF interaction with 14-3-3 protein[46]. As with the BRAF complex, no activation segment phosphorylation (Ser491, Ser494) was detected in the HSP90-CDC37-associated CRAF.

As PP5-dependent dephosphorylation of 14-3-3 binding sites was observed in both BRAF and CRAF, we considered the possibility that PP5 might play a role in modulating the regulatory interaction of these kinases with 14-3-3 proteins. We have previously shown that down-stream 'partner' proteins of HSP90-CDC37-bound kinases, such as CDK4 and CDK6, are able to extract the chaperone-bound kinase, presumably due to equilibrium exchange between the closed and open forms of the chaperone complex[47]. We, therefore, considered the possibility that 14-3-3 may be able to do this with HSP90-CDC37-bound BRAF^V600E so long as the key 14-3-3-interacting sites on BRAF^V600E were phosphorylated. Using an affinity 'pull-down' of the kinase in the purified HSP90–CDC37–BRAF^V600E complex in the presence of 14-3-3 protein in vitro, we observed co-precipitation of 14-3-3 protein

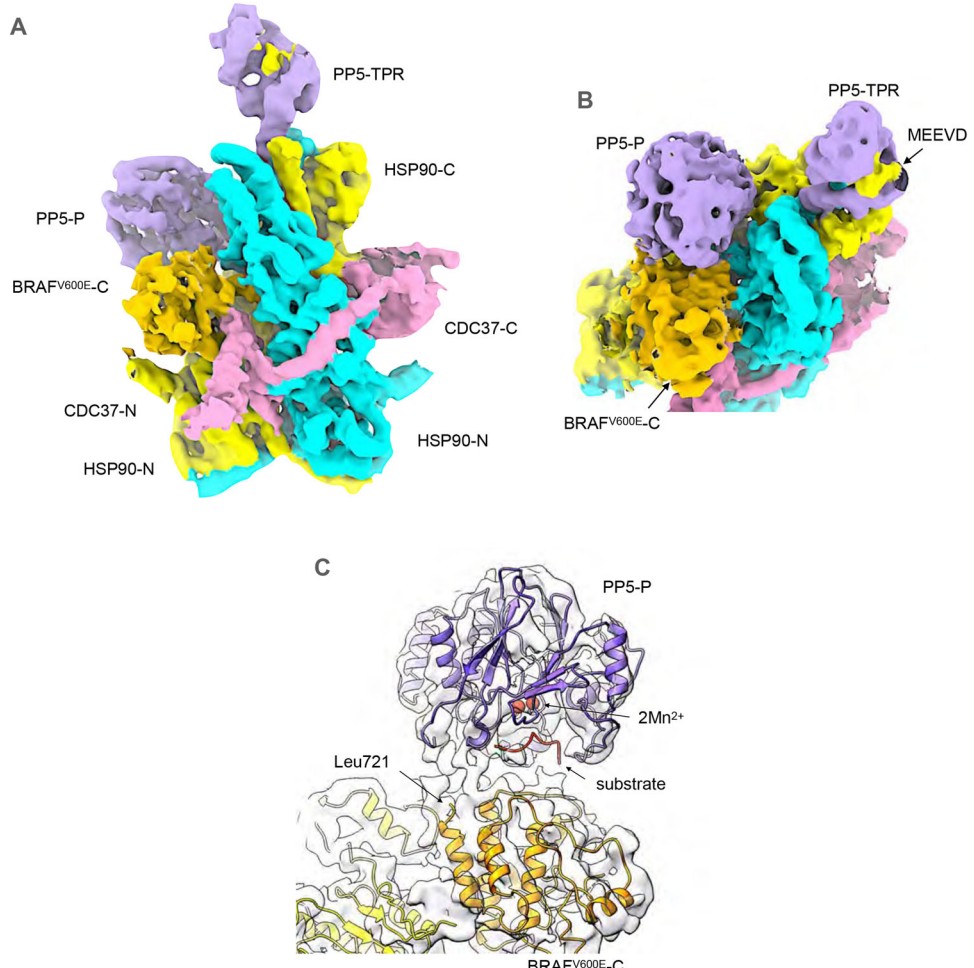

**Fig. 3 | Structure of HSP90–CDC37–BRAF^V600E complex with activated PP5.**
**A** Experimental cryoEM map of HSP90–CDC37–BRAF^V600E–PP5 complex, surface coloured as Fig. 2. The resolution of the region of the map corresponding to PP5 does not allow ab initio tracing of the polypeptide chain, which has therefore been modelled by rigid-body docking of previously determined crystal structures for the globular regions into the corresponding volume. The fit of the molecular model was then optimised by real-space refinement. **B** Like the autoinhibited complex, the cryoEM map suggests the presence of a bound C-terminal HSP90 MEEVD motif and the tip of the last helix of the TPR packs into the equivalent hydrophobic pocket in

the C-terminal domains of HSP90. However, the phosphatase domain of PP5 is no longer packed against its own TPR domain but is instead engaged with the ordered kinase C-lobe of BRAF^V600E. **C** PP5 phosphatase domain is docked against the kinase C-lobe of BRAF^V600E with its substrate-binding cleft facing the inter-helix loops. The end of the last α-helix in the BRAF^V600 C-lobe at Leu721 is well positioned for phosphorylated residues downstream (including Ser729—see Fig. 4.) to be engaged productively with the phosphatase active site. The positions of a bound substrate peptide and the catalytic manganese ions of PP5 are modelled from 5HPE[26].

(Fig. 5A). Western blotting confirmed that, consistent with the mass spectrometry observations, BRAF^V600E molecules within the purified HSP90–CDC37–BRAF^V600E sample retained phosphorylation on at least Ser729 which is required for high-affinity interaction with 14-3-3 (Fig. 5B). However, when PP5 was added to the HSP90–CDC37–BRAF^V600E complex, the amount of 14-3-3 protein co-precipitating with the BRAF^V600E was substantially diminished consistently with the dephosphorylation of BRAF-^V600E-Ser729 observed when PP5 was added. The isolated TPR domain of PP5 had no effect on the phosphorylation state of BRAF^V600E nor the level of 14-3-3 co-precipitated. We performed the same analysis with the HSP90–CDC37–CRAF complex, and again saw a substantial decrease in the amount of 14-3-3 that co-precipitated with CRAF following dephosphorylation of CRAF-Ser621 (Fig. 5C, D).

## Discussion

The structures presented here provide a view of a protein kinase other than CDK4[30] in a complex with CDC37 and HSP90 in the ATP-bound closed state[29]. This confirms a common mechanism of partial denaturation of the kinase domain, with the first strand of the

N-lobe linearised in the molecular clamp of the closed HSP90. While the remainder of the N-lobe of CDK4 was partially visible in some subsets of particles, the smaller and less structured N-lobe of BRAF is completely disordered in the HSP90–CDC37–BRAF^V600E complex. The molecular details of the interaction of the DFG motif of BRAF^V600E with CDC37 in the complex provide a satisfactory explanation for how the oncogenic mutation of Val600 within the early part of the activation loop, converts BRAF from a weakly dependent HSP90 client with only moderate affinity for CDC37[1], into a highly dependent client which is rapidly degraded when cells are treated with HSP90 inhibitors[24,48]. The potential importance of activation segment conformation in the client recognition process mediated by CDC37 is further underlined by the failure to detect any activation segment phosphorylation in either BRAF^V600E or CRAF in their purified complexes with HSP90-CDC37. This may indicate that activation segment phosphorylation only occurs after the release of the dephosphorylated kinase from the chaperone complex, or that its presence is inhibitory to recruitment by HSP90-CDC37—further work will be required to distinguish these possibilities.

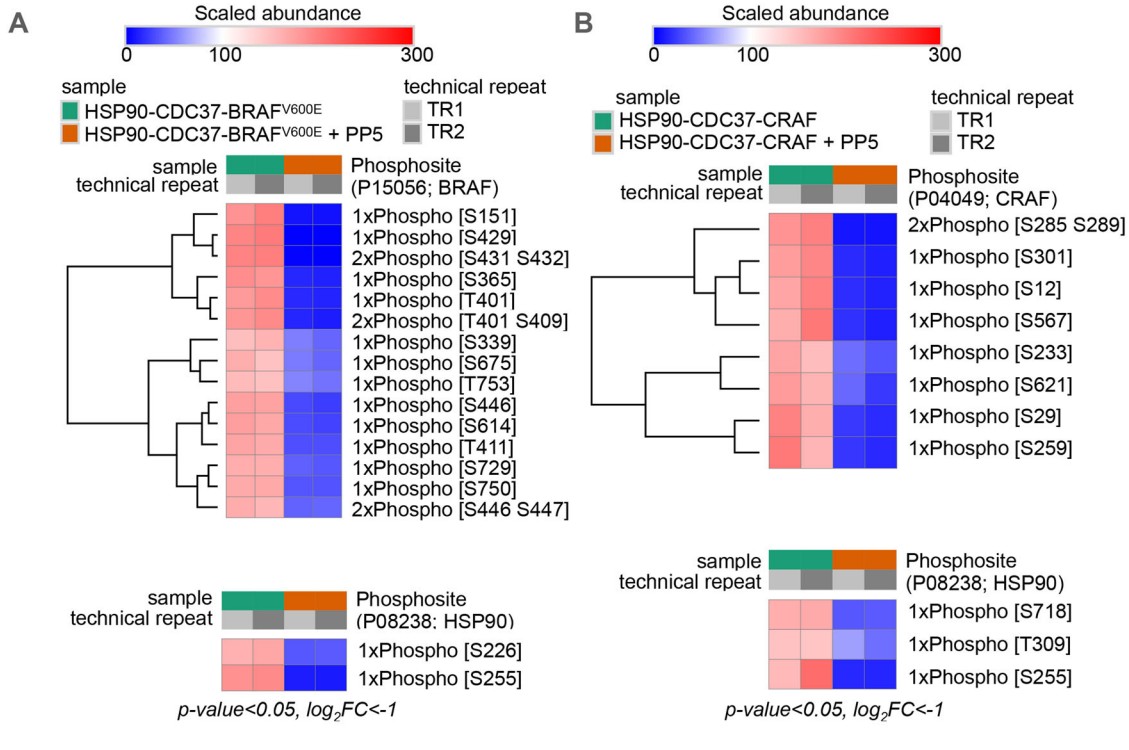

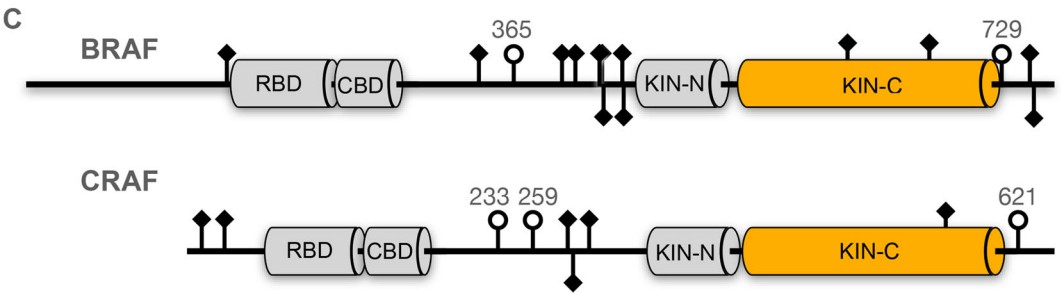

**Fig. 4 | PP5 is a comprehensive remover of client phosphorylations.**
**A** Comparison of phosphopeptide abundances from mass spectrometry analysis of BRAF[V600E] in the context of a complex with HSP90-CDC37, with or without incubation with PP5, each with two technical repeats utilising different labelling. Peptides are clustered on scaled abundance. Statistical analysis was performed with a two-sided *t*-test and multiple hypothesis testing corrections by permutation-based FDR. Significant hits were filtered for p-value<0.05, log2fold-change < −1 and had an FDR < 0.1. **B** As A. but for a CRAF–HSP90–CDC37 complex. **C** Schematic of phosphorylation sites identified in HSP90–CDC37-associated BRAF[V600E] and HSP90–CDC37-associated CRAF that are removed by the addition of PP5 (see Methods). Of the characterised globular regions, only the kinase C-lobe is ordered in complexes with HSP90−CDC37. Regulatory roles have been assigned in the literature for many of the sites identified in both proteins. Of particular interest are BRAF Ser365 and Ser729 and CRAF Ser233, Ser259 and Ser 621 (open circles) as they mediate the interaction of regulatory complexes with 14-3-3 proteins with the respective kinases[34,46]. Based on the relative position of the C-lobe and PP5-phosphatase, pSer729 is most likely to be the residue bound at the active site of the catalytically inactivated PP5 in the structure of the complex with HSP90−CDC37−BRAF[V600E].

PP5 docks onto HSP90−CDC37−BRAF[V600E] through a bipartite interaction mediated by the TPR domain, which binds the C-terminal MEEVD motif of HSP90 in its concave channel and plugs the tip of its terminal α-helix into one of two hydrophobic pockets formed at the interface of non-equivalent α-helices from each of the two HSP90 monomers. This mode of interaction is markedly different from that of FKBP51 with HSP90, which uses an N-terminal extension to its TPR to bind perpendicularly between the last helices of the HSP90 dimer[49]. We observe PP5 binding alternatively to both symmetry-related C-terminal pockets on HSP90, but with markedly different conformations depending on which side of the overall complex the phosphatase domain is positioned (Fig. 6A). When on the same face as the globular region of CDC37, which presents no phosphorylated substrate residues, the phosphatase domain remains associated with the TPR domain at the C-terminus

of the HSP90 dimer in an auto-inhibited conformation[37]. However, when bound with the phosphatase on the same face as the ordered C-lobe of the kinase, the phosphatase detaches from the TPR and docks against the middle domain of HSP90, with its substrate binding cleft in contact with the face of the kinase C-lobe from which the C-terminal segment extends, most likely held there by its interaction with one of the substrate phosphorylation sites that map to the early part of that segment (Fig. 6B, Supplementary movie 1). Considerable flexibility of the unstructured linker that connects the phosphatase to its TPR anchor would allow the phosphatase access to other substrate phosphorylation sites on the exposed surface of the C-lobe, and, indeed to parts of the kinase that are disordered in the complex, but nonetheless brought into general proximity to the phosphatase domain by their mutual binding to the HSP90−CDC37 'scaffold'.

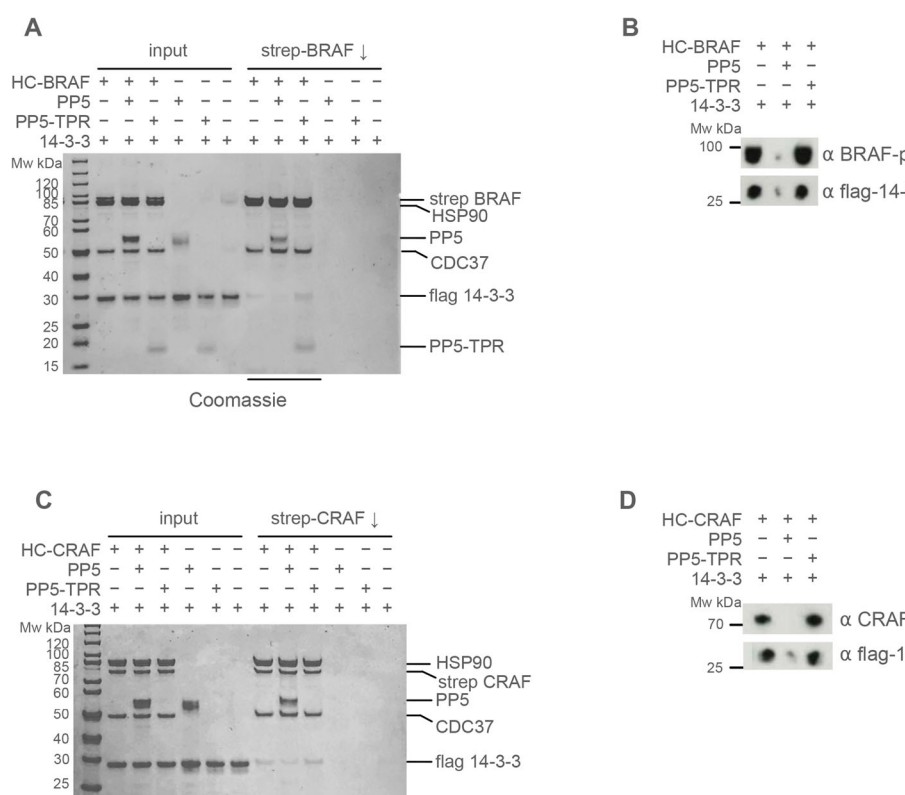

**Fig. 5 | Targeted dephosphorylation by PP5 regulates 14-3-3 binding to kinases.**
**A.** Purified HSP90–CDC37–BRAF[V600E] complex is able to co-precipitate flag-tagged 14-3-3 protein when the complex is pulled down on a strep-tag at the N-terminus of the kinase. The co-precipitated band is visible on a Coomassie-stained gel (underlined lanes), and very clearly in a Western blot developed with an antibody to 14-3-3. **B** Addition of PP5 substantially diminishes the levels of 14-3-3 co-precipitated, consistent with its ability to dephosphorylate the phosphorylation sites that mediate 14-3-3 interaction, whereas the PP5 TPR domain alone had no effect. The data shown are representative of >3 independent repeats. **B** Western blots of samples from the underlined lanes in (**A**). developed with antibodies to BRAF[V600E]-pSer729 (top) and flag-14-3-3 (bottom). Dephosphorylation of pSer729 substantially reduces the amount of 14-3-3 protein that associates with the BRAF[V600E]. **C** As **A** but for CRAF instead of BRAF[V600E]. The data shown are representative of >3 independent repeats. **D** As **B** but with the top blot developed with antibodies to CRAF-pSer621.

HSP90 sits at the heart of signal transduction within the eukaryotic cell[50], a substantial proportion of which is mediated by reversible phosphorylation. HSP90 in concert with its kinase-specific targeting partner CDC37 plays a critical role in the activation of the protein kinases that mediate this signalling, but the precise nature of that role remains obscure. Biochemical[4,47] and structural studies[30] (and see above) have clearly shown that interaction with the HSP90–CDC37 system results in catalytic silencing of a client kinase, through the partial unfolding of the kinase domain. However, association with HSP90–CDC37 also brings client kinases into the proximity of HSP90 co-chaperones that interact with the C-terminal MEEVD motif of the chaperone via their TPR domains. This exposes the client to modifications delivered by the catalytic domains of the TPR-cochaperones, which range from prolyl isomerases[49,51] to E3-ubiquitin ligases[52,53], and of most significance for protein kinase clients, a protein phosphatase - PP5.

Our data show that the overwhelming majority of Ser/Thr phosphorylations present on BRAF[V600E] or CRAF bound to HSP90–CDC37, are removed by PP5. Thus, PP5 effectively provides a 'factory reset' of the client kinase by removing whatever regulatory modifications may have been applied to it before it bound to HSP90–CDC37, on both N- and C-terminal sides of the kinase domain that drive chaperone recruitment (Fig. 6C). In the case of CRAF and BRAF[V600E], chaperone-targeted PP5 activity removes phosphorylation sites required for their regulatory interaction with 14-3-3 proteins, and could therefore exert a direct effect on the activity of the client post-release. We have here demonstrated chaperone-targeted kinase dephosphorylation by PP5 for two different but related kinases—BRAF and CRAF. However, given the similarity of the way kinases as different as BRAF and CDK4 have been shown to interact with HSP90–CDC37, it is reasonable to suppose that dephosphorylation by PP5 may be a common experience for all HSP90 client kinases. Together with its ability to also remove the phosphorylation of CDC37 and thereby destabilise the association of the kinase client with the chaperone complex[23], PP5 could then provide directionality to the interaction of the kinase with the chaperone complex, ensuring the release of the client from the HSP90–CDC37 platform as a *tabula rasa*, ready for whatever new phosphorylation events are required for its regulated function in the cell.

## Methods

### Protein expression and purification

Full-length human HSP90β, CDC37 and BRAF[V600E] were subcloned into the baculovirus vector pBIG1a[54] with an N-terminal His$_8$ tag on HSP90β, a C-terminal His$_8$ on CDC37 and N-terminal His$_8$-2xStrep tag on the BRAF[V600E]. Human rhinovirus 3C protease recognition sites were introduced between the proteins and the fusion tags. An identical approach was used for the expression of HSP90-CDC37-CRAF.

*Sf9* cells (Invitrogen) were transfected with 1 μg of pBIG1a HSP90β, CDC37 and BRAF[V600E] for viral production. For protein expression,

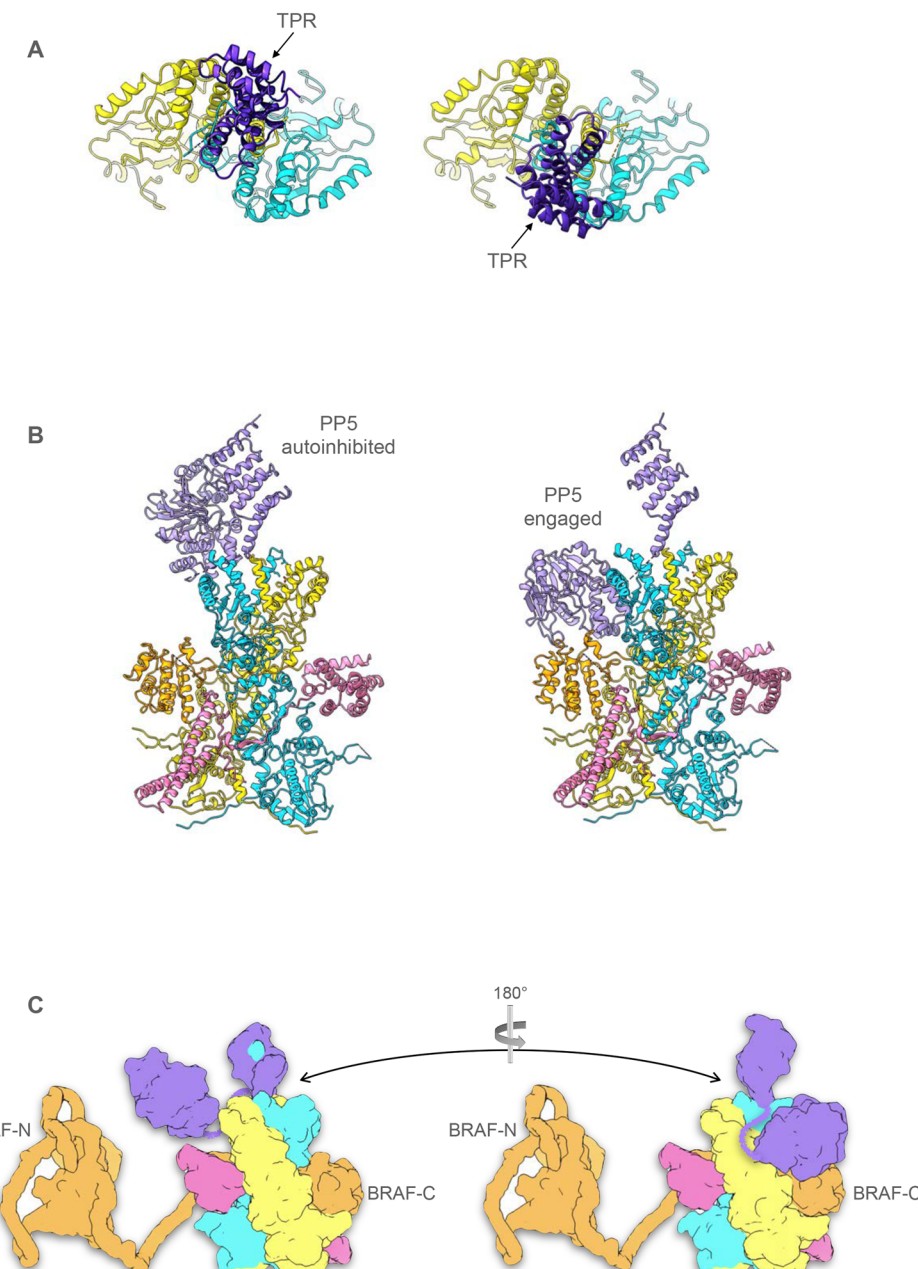

**Fig. 6 | Conformational rearrangements of PP5 facilitate dephosphorylation. A** The HSP90 dimer provides two symmetry equivalent alternative and mutually exclusive binding sites for the TPR domain of PP5, which are both seen in the complexes described here, and directs the associated phosphatase domain to different faces of the complex. **B** Following binding to either of the two symmetry equivalent binding sites at the C-terminus of HSP90, the phosphatase domain of PP5 can release from its autoinhibited interaction with its own TPR domain (left), and engage with substrate phosphorylation sites on the chaperone components or the bound client (right). The model shown on the right is based on the experimental

'closed' structure (see above), but with the PP5 component rotated to the same side as the phosphatase in the experimental 'open' structure by the superimposition of the TPR domains. A movie illustrating the conformational switch between the 'closed' catalytically autoinhibited conformation and the 'open' substrate-engaged conformation is provided in Supplementary Movie 1. **C** By switching PP5 binding between the two TPR-binding sites on the HSP90 dimer, the flexibly attached phosphatase domain can access phosphorylation sites upstream and downstream of the core interacting domain of the kinase client, taking advantage of the partly unfolded and 'linearised' state that binding to HSP90–CDC37 facilitates.

*Sf*9 cells were infected with HSP90β, CDC37 and BRAF[V600E] baculovirus at an MOI of 2 and incubated for 72 h at 26 °C.

 *Sf*9 cells were lysed and sonicated in 40 mM Hepes pH 7.4, 150 mM NaCl, 10 mM KCl, 20 mM $Na_2MoO_4$, 20 mM imidazole, 0.5 mM TCEP, 10% glycerol, 2U/ml Turbo DNAase (Invitrogen), EDTA-free protease inhibitor cocktail tablets and phosphatase inhibitor tablets (Roche). The NaCl concentration was increased to 750 mM before incubating the lysate with talon resin (Takara Bio) for 2 hours at 4 °C. The resin was washed sequentially with lysis buffer

containing 750-600-450-300 and 150 mM NaCl. The protein complex was eluted from the resin in 40 mM Hepes pH 7.5, 100 mM NaCl, 10 mM KCl, 20 mM $Na_2MoO_4$, 500 mM imidazole, 0.5 mM TCEP, 10% glycerol. The eluate from the Talon resin was applied to a 2 ml streptactin column (IBA) in Streptactin binding buffer consisting of 40 mM Hepes pH 7.4, 150 mM NaCl, 10 mM KCl, 10 mM MgCl2, 20 mM $Na_2MoO_4$, 0.5 mM TCEP, 10% glycerol and eluted with 75 mM Biotin. Elutions from the Streptactin column were applied to a Superdex S200 26/60 size exclusion column (GE Healthcare) and

eluted in Streptactin binding buffer. An identical approach was used for the expression of HSP90-CDC37-CRAF.

Human PP5 residues 16-499 with or without a D274N mutation were cloned into pGEX6P1 with an N-terminal GST tag and C-terminal His$_6$ tag. PP5 was expressed in *Escherichia coli* and purified as previously described[37]. Human PP5 TPR residues 19-147 subcloned into the pET28a vector with an N-terminal His$_6$ tag was expressed in *E. coli*. The cells were lysed and sonicated in 40 mM Hepes pH 8.0, 300 mM NaCl, 5% glycerol, 0.5 mM TCEP and EDTA-free protease inhibitor cocktail tablets. The lysate was incubated with talon resin (Takara Bio) for 1 hour at 4 °C. The resin was washed sequentially with lysis buffer, and the protein was eluted in 40 mM Hepes pH 8.0, 300 mM NaCl, 250 mM imidazole, 0.5 mM TCEP, and 5% glycerol. The eluate from the talon resin was applied to a Superdex S75 26/60 size exclusion column (GE Healthcare) in 25 mM Hepes pH 8, 150 mM NaCl, 0.5 mM TCEP and 5% glycerol.

The ζ-isoform of 14-3-3 with an N-terminal 6xHis-FLAG tag was cloned into pRSET-A and expressed in *E. coli* BL21 DE3 cells. Cells were lysed in 40 mM Tris pH 8, 0.5 mM TCEP, 500 mM NaCl, 5 mM imidazole and EDTA-free protease inhibitor cocktail tablets (Roche). The extract was purified with a HiTrap Talon Crude column and eluted in lysis buffer with 500 mM imidazole. The elution was applied to a HiPrep Desalting column, followed by ion exchange chromatography on a HiTrap Q and size exclusion on a HiLoad Superdex 75 in 20 mM Hepes pH 7.4, 0.5 mM TCEP, 50 mM NaCl 10% glycerol (all chromatography media from Cytiva/GE).

### HSP90–CDC37–BRAF$^{V600E}$–PP5 complex assembly

To assemble the PP5 complex for CryoEM, the HSP90–CDC37–BRAF$^{V600E}$ complex was purified as described above, but after the complex was eluted from the Strep-tactin column, the Na$_2$MoO$_4$ and biotin were removed from the buffer by buffer exchanging using a 100 kDa Mwt cut-off concentrator. This sample was then incubated with a 2× molar excess of PP5 for 2 h at 4 °C. The sample was loaded onto a Superdex s200 10/300 size exclusion column (GE Healthcare) and eluted in 100 mM NaCl, 25 mM Hepes, 10 mM KCl and 1 mM MnCl$_2$, 0.2 mM TCEP and 2% glycerol. Samples prepared for CryoEM were further crosslinked with 1 mM BS3 (Fisher Scientific UK Ltd.) for 30 minutes at room temperature and quenched with 20 mM Tris pH 7.5. The purification and activity of the complexes are shown in Supplementary Fig. 1.

### CryoEM grid preparation data collection

Prior to grid preparation, the crosslinked HSP90–CDC37–BRAF$^{V600E}$–PP5 complex was concentrated to 1 μM and 3 μl of the sample was applied to carbon grids (Quantifoil R1.2/1.3, Cu, 300 mesh) which were glow discharged using a Tergeo Plasma Cleaner (Pie Scientific). The sample was blotted for 5 s using a Leica EM GP2 (Leica microsystems) and plunge-frozen in a 37% liquid ethane/propane mixture.

Three different datasets were collected using a Titan Krios (Thermo Fisher Scientific) equipped with a Falcon 4 camera in counting mode, at a magnification of 96,000, which corresponds to a pixel size of 0.86 Å/pixel. EPU software version 2.11 (Thermo Fisher Scientific) was used to collect data with a defocus range of −2.5 to −1.3 μm at a dose rate of 9.5 e$^-$/Å$^2$/s, for a total exposure of 4.7 s and with 56 frames resulting in a total dose of 45 e$^-$/Å$^2$. Examples of cryoEM images and class averages are shown in Supplementary Fig. 1.

### CryoEM data processing

Movies of images from the three datasets were motion-corrected separately using MotionCor2[55] and were binned to 1.72 Å/pixel. A total of 21,196 micrographs were collected from the three different datasets. CTF estimation, particle picking and the first round of reference-free 2D classification were carried out on each dataset separately initially. CTF was estimated using Patch CTF in

cryoSPARC2 v3.3.1[56]. About 500 particles were manually picked initially to generate 2D templates for picking using Topaz[57]. One round of reference-free 2D classification of particles picked using Topaz was performed in CryoSPARC to remove noisy class averages. At this stage, the particles from the first round of 2D classification from each dataset were combined, and the second round of 2D classification was performed on the combined total of 721,941 particles. Class averages with high-resolution features were selected after this round of 2D classification, and three Ab initio models were generated from 537,780 particles using CryoSPARC.

All subsequent processing steps were done in RELION v3.1.2[58]. An initial round of 3D classification was performed using the particles and the ab-initio model obtained in CryoSPARC. Three classes had clearly recognisable density for HSP90, CDC37, BRAF and PP5 and were subjected to the second round of 3D classification. After this round, three distinct classes were observed, one containing only HSP90, CDC37 and BRAF, one containing HSP90, CDC37, BRAF and PP5, which is a more closed conformation bound to the C-terminal of HSP90 and the third containing HSP90, CDC37, BRAF and PP5 which is opened up and the TPR domain is bound to the C-terminal of HSP90 but where the phosphatase is engaged in the middle domain region of HSP90. To use the best signal from the HSP90–CDC37–BRAF complex class for particle polishing, particles from the three classes were combined (533,127 particles in total) and refined to 3.6 Å resolution. Only HSP90–CDC37–BRAF density was visible in this class. Particle polishing and CTF refinement were performed on these particles, followed by 3D refinement. A round of 3D classification was performed to retrieve back the three classes, which now contain polished particles. The overall resolution for all three classes improved after 3D refinement.

To improve the resolution of the PP5 domains, the particles from the two classes which contained PP5 were further classified using signal subtraction and focused 3D classification without alignments, with masks around the PP5 domains. After reverting to the original particles and applying a soft mask around the whole complex, the best-focused class for the HSP90, CDC37, BRAF and PP5 (in a more 'closed' conformation) refined to a resolution of 3.9 Å and the best class for the HSP90, CDC37, BRAF and PP5 (in an open conformation) refined to 4.2 Å, with improved density observed for the PP5 domains in both classes. Particles from all three classes were re-extracted at 0.86 A/pix, and a final round of particle polishing was performed on all three classes individually, followed by postprocessing in RELION in which the nominal resolution was determined by the gold standard Fourier shell correlation (FSC) method[59]. Maps were subsequently postprocessed using DeepEMhancer[60].

The data processing flow is shown in Supplementary Fig. 2, local resolution variations, and FSCcurves for refinement and model fitting are shown in Supplementary Fig. 3., and 3D plots of particle orientations are shown in Supplementary Fig. 4.

### Model building

Atomic models were derived from the cryoEM structure of an HSP90-CDC37-CDK4 complex (PDB code: 5FWK), and crystal structures of PP5 protein and domains (PDB codes: 1WAO, 5HPE and 1A17) and BRAF kinase domain (PDB code: 1UWH) docked as rigid bodies into experimental volumes using ChimeraX[61]. The local fit of the models was adjusted manually in Coot[62], and the global fit was optimised using Phenix.refine[63]. Parameters defining the data collection and the quality of the final atomic models (PDB codes: 7ZR0, 7ZR6, 7ZR5) are given in Table 1.

### Dephosphorylation assays

To monitor dephosphorylation by PP5, 0.15 μM of HSP90–CDC37–BRAF$^{V600E}$ complex or HSP90–CDC37–CRAF complex was mixed with 0.3 μM of PP5 in a buffer containing 100 mM NaCl, 25 mM Hepes pH 8,

**Table 1 | CryoEM data collection and model refinement parameters for HSP90–CDC37–BRAFV600E (HCK), HSP90–CDC37–BRAFV600E–PP5 open (HCKPo) and HSP90–CDC37–BRAFV600E–PP5 closed (HCKPc)**

| | HCK | HCKPo | HCKPc |
|---|---|---|---|
| Magnification | **96,000** | | |
| Voltage (kV) | 300 | | |
| Electron exposure (e–/Å²) | 45.0 | | |
| Defocus range (μm) | −1.3 to −2.5 | | |
| Pixel size (Å) | 0.86 | | |
| Symmetry imposed | C1 | | |
| Initial particle images (no.) | 1,126,606 (before the split into three models) | | |
| | **HCK** | **HCKPo** | **HCKPc** |
| Final particle images (no.) | 400,624 | 67,985 | 105,063 |
| Map resolution (Å) | 3.4 | 4.2 | 3.9 |
| FSC threshold | 0.143 | 0.143 | 0.143 |
| **Refinement** | | | |
| Initial model used (PDB code) | 5FWK, 1UWH | 5FWK, 1UWH, 1WAO, 5HPE, 1A17 | 5FWK, 1UWH, 1WAO, 5HPE, 1A17 |
| Map resolution range (Å) | 3.29–6.9 | 3.68–10.39 | 4.02–11.75 |
| Map sharpening B factor (Å) | −130 | −84 | −180 |
| Non-hydrogen atoms | 14052 | 17742 | 17560 |
| Protein residues | 1716 | 2173 | 2150 |
| Co-factor residues | 2 | 2 | 2 |
| **B-factors (Å²)** | | | |
| Protein | 61.6 | 115.2 | 90.0 |
| Co-factor | 48.6 | 60.2 | 41.8 |
| **R.m.s. deviations** | | | |
| Bond lengths (Å) | 0.007 | 0.003 | 0.009 |
| Bond angles (°) | 0.995 | 0.674 | 1.017 |
| **Validation** | | | |
| MolProbity Score | 2.1 | 2.0 | 2.1 |
| Clashscore | 10.5 | 10.1 | 9.6 |
| Poor rotamers (%) | 0.0 | 0.1 | 0.3 |
| **Ramachandran plot** | | | |
| Favoured (%) | 89.3 | 92.0 | 90.3 |
| Allowed (%) | 10.1 | 7.8 | 9.1 |
| Disallowed (%) | 0.5 | 0.2 | 0.6 |
| **Model fit** | | | |
| CC mask | 0.78 | 0.66 | 0.70 |
| CC volume | 0.80 | 0.67 | 0.72 |
| Map-model FSC 0,0.143,0.5 | 2.1,2.2,3.4 | 1.8,2.3,4.2 | 1.7,2.1,3.9 |

10 mM KCl, 1 mM MnCl2, 0.2 mM TCEP, 2% glycerol and 2.5 mM MgCl$_2$. The reaction was started by incubating the samples at 30 °C. Samples were taken over 45 min for SDS-PAGE analysis. Phosphorylation-site specific antibodies were used in western blots to probe the phosphorylation states of phospho-Ser13 Cdc37 (MA533209 Invitrogen) diluted 1/2500, phospho-Ser729 BRAF (ab124794 Abcam) diluted 1/1000 and phospho-Ser621 CRAF (ab157201 Abcam) diluted 1/1000. Uncropped raw images for this and all other western blots presented are shown in Supplementary Fig. 5.

**In vitro pull-down assays**
After performing the PP5 dephosphorylation assays as described above, the samples were incubated with 0.6 μM 14-3-3 for 30 min. This sample was then mixed with 50 μl of Streptactin resin (IBA) for a further 30 min. The flow-through was removed and the resin was washed with 3 × 100 μl of dephosphorylation assay buffer. The complex was eluted in dephosphorylation assay buffer containing 2.5 mM desthiobiotin, followed by SDS-PAGE analysis and western blot. Western blots utilised phospho-Ser729 BRAF and phospho-Ser621 CRAF antibodies as above, and an antibody to 14-3-3 (#8312 Cell Signalling) diluted 1/1000.

**Mass spectrometry phosphorylation analysis**
Samples were split into two equal parts and diluted up to 100 μL with 100 mM triethylammonium bicarbonate (TEAB) followed by one-step reduction/alkylation with 5 mM TCEP and 10 mM iodoacetamide for 45 min at room temperature. Proteins were then digested overnight with 50 ng/μL trypsin (Pierce). Peptides were labelled with the TMT-10plex reagents (four labels used) according to the manufacturer's instructions (Thermo), followed by C18 clean-up using the Pierce Peptide Desalting Spin Columns. Phosphopeptides were enriched with the High-Select™ Fe-NTA Phosphopeptide Enrichment Kit (Thermo). Both the enrichment eluent and flowthrough (FT) were further subjected to mass spectrometry analysis.

LC-MS analysis was performed on the Dionex UltiMate 3000 UHPLC system coupled with the Orbitrap Lumos Mass Spectrometer (Thermo Scientific). Each sample was reconstituted in 30 μL 0.1% formic acid, and 15 μL were loaded to the Acclaim PepMap 100, 100 μm × 2 cm C18, 5 μm trapping column at 10 μL/min flow rate of 0.1% formic acid loading buffer. Peptides were analysed with an Acclaim PepMap (75 μm × 50 cm, 2 μm, 100 Å) C18 capillary column connected to a stainless-steel emitter with integrated liquid junction (cat# PSSELJ, MSWIL) fitted on a PSS2 adaptor (MSWIL) on the EASY-Spray source at 45 °C. Mobile phase A was 0.1% formic acid, and mobile phase B was 80% acetonitrile, 0.1% formic acid. The gradient separation method at a flow rate of 300 nL/min was the following: for 65 min (or 95 min for FT) gradient from 5%–38% B, for 5 min up to 95% B, for 5 min isocratic at 95% B, re-equilibration to 5% B in 5 min, for 10 min isocratic at 5% B. Each sample was injected twice. Precursors between 375 and 1500 m/z were selected at 120,000 resolution in the top speed mode in 3 s and were isolated for HCD fragmentation (collision energy 38%) with quadrupole isolation width 0.7 Th, Orbitrap detection at 50,000 resolution (or 30,000 for FT sample), max IT 100 ms (or 50 ms for FT sample) and AGC $1 \times 10^5$. Targeted MS precursors were dynamically excluded for further isolation and activation for 30 or 45 s with 7 ppm mass tolerance.

The raw files were processed in Proteome Discoverer 2.4 (Thermo Scientific) with the SequestHT search engine for peptide identification and quantification. The precursor and fragment ion mass tolerances were 20 ppm and 0.02 Da, respectively. Spectra were searched for fully tryptic peptides with a maximum of 2 missed cleavages and a minimum length of 6 amino acids. TMT6plex at N-terminus/K and Carbamidomethyl at C were selected as static modifications. Oxidation of M, Deamidation of N/Q and Phosphorylation of S/T/Y were selected as dynamic modifications. Spectra were searched against reviewed UniProt Homo sapiens protein entries (downloaded 18/01/2022), peptide confidence was estimated with the Percolator node, and peptides were filtered at $q$-value < 0.01 based on decoy database search. Results were reported at the peptide level, and at least two unique peptides were required for the identification of the proteins of interest. The reporter ion quantifier node included a TMT quantification method with an integration window tolerance of 15 ppm. Only peptides with average reporter signal-to-noise >3 were used, and phosphorylation localisation probabilities were estimated with the IMP-ptmRS node with an initial minimum probability of 25%. Selected phosphosites had a minimum localisation probability of 98%. Statistical analysis was performed in Perseus software using a two-sided $t$-test and multiple hypothesis testing corrections by permutation-based FDR. Significant hits were filtered for $p$-value < 0.05, log2fold-change < −1 and had an FDR < 0.1.

## Reporting summary

Further information on research design is available in the Nature Portfolio Reporting Summary linked to this article.

## Data availability

The cryoEM data generated in this study have been deposited in the PDB and EMDB under PDB ID 7ZR0 and EMD-14875 (HCK); PDB ID 7ZR6 and EMD-14884 (HCKPo); PDB ID 7ZR5 and EMD-14883 (HCKPc). Proteomic data generated in this study are available from the PRIDE database and the accession codes PXD033678 (HSP90–CDC37–BRAF complex) and PXD035934 (HSP90-CDC37-CRAF complex). Previously published structures used for the interpretation of the cryoEM volumes generated in this study are available in the PDB under HSP90-CDC37-CDK4 complex–PDB ID 5FWK; PP5 protein and domains–PDB IDs 1WAO, 5HPE, 1A17 and 2BUG; BRAF kinase domain–PDB ID 1UWH. Uncropped western blots are provided in Supplementary Fig. 5.

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

## Acknowledgements

We thank Rebecca Thompson, Emma Hesketh and Louie Aspinall for assistance with cryoEM data collection at The University of Leeds, Fabienne Beuron for assistance and advice regarding grid preparation, Lihong Zhou for assistance with insect cell expression, Basil Greber for advice on image processing, Antony Oliver for assistance with data handling, and Cara Vaughan and Chris Prodromou for contributions at earlier stages of the work. EM facilities at the Astbury Biostructure Laboratory at the University of Leeds are funded by the University of Leeds ABSL award and Wellcome Trust award 108466/Z/15/Z. EM facilities at Sussex University are funded by Wellcome Trust Award Enhancement Grant 095605/Z/11/A —to L.H.P.—and the RM Phillips Trust). This work was supported by a Wellcome Trust Investigator Award 210719/Z/18/Z (L.H.P.) and Cancer Research UK Centre Grant C309/A25144 (J.S.C and T.I.R.).

## Author contributions

Conceptualisation: J.O. and L.H.P.; Methodology: J.O., X.A.G., E.A.O., P.S., T.I.R., J.S.C., and L.H.P.; Validation: J.O., J.S.C., and L.H.P.; Formal analysis: J.O., T.I.R., J.S.C., and L.H.P.; Investigation: All Authors; Writing —original draft: L.H.P.; Writing—review and editing: All authors; Visualisation L.H.P.; Supervision: L.H.P.; Funding Acquisition: L.H.P.

## Competing interests

The authors declare no competing interests.
