## [Peer Review File · Nature Communications]

REVIEWER COMMENTS

Reviewer #1 (Remarks to the Author):

In this manuscript, Oberoi et al. solved the cryo-EM structures of HSP90-Cdc37-BRAFV600E complex, in the absence or presence of PP5, and demonstrated the ability of PP5 in dephosphorylating Ser/Thr residues in HSP90 and BRAFV600E in the complex. The study provided important molecular insights into the role of PP5 in resetting HSP90 and its kinase clients in the HSP90 chaperone cycle. While only the C-terminal lobe of BRAFV600E is structured in the complex, there is so far only another structure for the HSP90-Cdc37-protein kinase (CDK4) ternary complex. Hence, the study also advances the HSP90 field by offering structural insights into the interaction between HSP90-Cdc37 and kinase clients. Overall, this is a significant study, and the technical quality of the work is excellent. I recommend the acceptance of the manuscript for publication in Nat. Commun. after some revisions:

1. The authors provided phosphoproteomic data to support the capability of PP5 in removing phosphate groups from 2 sites in HSP90, and 12 sites in BRAFV600E in the purified HSP90-Cdc37-BRAFV600E complex in vitro. This part of the study can be strengthened if cellular data supporting this role of PP5 are furnished. For instance, the authors could examine how genetic depletion and overexpression of PP5 in melanoma cells carrying the BRAFV600E mutation modulate the levels of phosphorylation at aforementioned sites in HSP90 and BRAFV600E. In addition, the authors should provide, in Supplementary Figures, the annotated MS/MS in supporting the identification of phosphosites in HSP90 and BRAFV600E.
2. In the phosphoproteomic data presented in Supplementary Table, the authors observed elevated phosphorylation of several sites in PP5 after PP5 addition, where appreciable signals for phosphopeptides were also detected for PP5 in the HCB sample without the addition of PP5. Does this suggest that the HCB complex carries PP5 contamination? Since the proteins were purified from insect cells, these results were somewhat surprising.
3. Multiple bands were detected for the HSP90-Cdc37-BRAFV600E-PP5 complex after BS3 crosslinking (Supplementary Figure 1). Can the authors comment on the heterogeneity of the complex?

Reviewer #2 (Remarks to the Author):

This manuscript describes cryo-EM structures of HSP90 in complex with the co-chaperone CDC37, protein kinase BRAF V600E and phosphatase PP5 with PP5 in two conformations, and a structure of the HSP90/CDC37/ BRAF V600E complex. In addition, the authors carried out by mass spectrometry experiments to map the potential sites on the different components of the complex that could be de-phosphorylated by PP5. The structures presented here could potentially help to elaborate the mechanism of HSP90, CDC37 or BRAF dephosphorylation by PP5, however the results and proposed mechanism are not supported by any in vitro biological assays, or directly evidenced in the cryo-EM structures – in general, the title and conclusions reached in the manuscript tend to overstate what can be inferred from the data presented. The manuscript would also benefit from extensive editing to improve readability and to improve the clarity of the figures. Overall, the results and structures presented in this manuscript are of general interest to those in the kinase/Ras/RAF signaling field, and may be suitable for publication in Nature Communications, once the following major concerns have been addressed.

Major comments:

- The manuscript tends to overstate the results, or the description of the results is not consistent with or fully supported by the figures. To fully support the statement made by the title ('forms a structural platform for kinase dephosphorylation') and the model presented in Figure 4, additional evidence is needed – either biochemical evidence or further structural evidence. The data presented provides a snapshot of the likely catalytic cycle and does not fully elucidate the dephosphorylation mechanism of PP5. Alternatively, the title and abstract can be changed to better reflect the data presented in the current manuscript. The current structures only show the inactive PP5 binds to the system without engaging the phosphorylated sites in the components.
- The V600E of BRAF mutant was used throughout the paper. It is briefly discussed that this mutant was used as it appears to be highly HSP90 dependent. Please elaborate, and please discuss how the structure may differ with wild-type BRAF in terms of binding to HSP90.
- The fact that the complex likely is dynamic was referred to repeatedly in the text. Have the authors considered using multi-body refinement (in Relion 4) or 3D variability analysis (in Cryosparc) or 3D classification to better characterize the dynamic states of the complex?
- Interfaces are described for HSP90/CDC37, CDC37/BRAF, but not PP5. Please describe any interfaces between PP5 and other proteins in the complex, and also describe the PP5 interactions in the context of the mass spectrometry data.
- Please include legends for all Supplementary Tables and Supplementary Figures. Please remove all erroneous text on the Supplementary Figures and please include it in the figure legends.

- The manuscript will benefit from an initial figure which presents the overall EM maps and structures alongside each other to better orient the reader.
- Mass spectrometry data is provided that indicate potential dephosphorylation sites for HSP90 and BRAF by PP5. Results are only shown for BRAF in the supplementary figure, and no results are shown for HSP90. Please include the results for HSP90 as a figure, and discuss the results closely considering the structural data – are the potential phosphorylation sites phosphorylated in the structures? Given that the mass spectrometry data represents a significant amount of work, please consider including them as a main figure.
- The writing in the manuscript can be somewhat unclear and would benefit from extensive editing to ensure readability and clarity. Please see below minor comments for detailed suggestions. More thorough and more careful labelling of the figures will aid the reader in interpreting the results.

Minor comments:

- A preprint has been released describing the structure of a Raf-1(CRAF)/HSP90/CDK37 complex (doi: 10.1101/2022.05.04.490607). The structures presented should be compared with the structures presented in this manuscript briefly in the discussion.
- There is a lot of white space in the figures – please consider arranging the figures to reduce the white space.
- In the manuscript, the terms ‘closed’ and ‘open’ are used interchangeably, referring to both the PP5 and HSP90. This is confusing for the reader, and it would be clearer if different terms are used for the two conformational states of the HSP90/CDK37/BRAF/PP5 complex.
- Likewise, when terms like ‘active’ or ‘autoinhibited’ are used, it should be explicit as to if the term is referring to the whole HSP90 complex, or a specific component of the complex.
- In Supplementary Table 1, please include a map/model FSC value, and include real-space map fitting statistics (i.e. CCmask, CCvolume). In Supplementary Figure 2, please be explicit about which ab initio models were used to generate the high-resolution models. Please also note that ‘CTF’ should be in all-caps. Please add scale bars for the micrograph and 2D classes in Supplementary Figure 2.
- In Supplementary Figure 3, please re-plot the FSC curves in a suitable plotting program so they are clearer to the reader. Please also include a model-map FSC curve and indicate the FSC=0.5 and 0.143 thresholds in the plot.
- Figures are labelled interchangeably as ‘Figure’ and ‘Fig’. Please be consistent.
- Page 6, line 1 – it is unclear which part of the complex Ser 13 belongs to.
- Page 10, line 18-19 – please note typo ‘While the remainder of the N-lobe of CDK4 was partially visible in some subsets of...’.

Reviewer #3 (Remarks to the Author):

The manuscript presented by Oberoi et al shows an interesting structural characterization of the mechanism that allows dephosphorylation of BRAF kinase by PP5 with the necessary contribution of Hsp90 and its cochaperone Cdc37. The authors provide three snapshots of the Hsp90-Cdc37-BRAF complex: with PP5 in an autoinhibited and active conformations, and without PP5. The structural results are complemented by functional assays in which the activity of PP5 is assessed by an exhaustive proteomic analysis of the phosphorylation levels of the proteins that form the complex. The significant dephosphorylation observed confirms the activation of PP5 by binding to Hsp90-Cdc37 complex, which results in the inactivation of the oncogenic kinase BRAF.

The results presented here are solidly supported and consistent with previous evidence. The manuscript is well discussed and easy to follow. However, there are two main concerns for me. One of them is that some regions in the CryoEM structures display a limited resolution, which doesn't allow a proper modelling of some key elements like the Hsp90 C-terminal MEEVD. The second one is that the validation reports show a poor fitting of a large portion of Cdc37 and PP5 in both conformations, indicating a lack of density to accommodate these proteins. The authors tackled these issues by docking previously described atomic structures, which is a nice approach to support their proposed mechanism, but can lead to an overinterpretation of the data. I would suggest to clarify that these are tentative models that cannot be fully modelled based on the CryoEM data, and remove these regions from the deposited structures. Alternatively, some other experiments could reinforce the proposed models. For instance, XL-MS could confirm interacting regions in medium-resolution regions.

There are some other issues that should be addressed:

-The title suggests that the mechanism described is shared among a variety of kinases; however, there is no evidence that supports this. In my opinion, it should be rephrased to a more precise title that describes the particular complex studied here.

-Introduction: When BRAFV600E mutant is introduced, it is stated to be highly dependent on Hsp90. A reference should be provided for this.

-Supplementary figure 1B shows a biochemical analysis of the complex among the four proteins, with a well-defined peak. It is surprising that in the 3D classification a large proportion of the

particles (around 63%) are not bound to PP5, especially when the complex was crosslinked with BS3. This is highly unexpected, and the authors should provide a suitable explanation for this.

-Another surprising aspect of the purification of the complex is that two conformations of PP5 are observed, but only one conformation (ATP-closed) is found for Hsp90. Why can't other conformations be found in the sample? Is this due to the presence of molybdate or is it related with the complex formation?

-General comments for all the figures: I would recommend showing equivalent views of all the complexes, including at least two orthogonal views to visualize all the features of the maps, and indicating the rotation. An image showing the atomic models docked into the final maps would be desirable to assess the quality of the models.

-Figure 1B: The nucleotide would be better visualized if it were depicted in a different colour. Figure 1C: Ser13 should be highlighted somehow.

-Page 6, lines 1-3: When speaking about Figure 1C: "Ser13... is clearly phosphorylated within the complex and engaged ^{[[1]]}_{[[SEP]]} with the side chains of CDC37 residues His33 and Arg36, and Lys406 of HSP90". This phosphorylation and its interactions with adjacent residues were already structurally described in Verba et al. (2016), and this should be clearly stated in the text.

-Figure 2B and C: The authors acknowledge that the resolution is insufficient to directly model the C-terminal MEEVD sequence of Hsp90. However, they built an atomic model based on a previous structure. The experimental data doesn't provide any evidence that this region should be modelled as it has been. The text should clearly indicate that this is a tentative model and these residues should be removed from the deposited structure. This should be taken into account for Figure 3 too.

-Page 8 (final paragraph): The conformational change on PP5 would be better visualized if supported by a figure. It could be included as part of figure 4 or a modified version of Figure 4C.

-When doing the image processing, signal subtraction and focused 3D classification without alignment were used to try to improve PP5 resolution. Other tools such as Relion multi-body refinement or cryoSPARC local refinement protocol are very useful approaches for these tasks. Have the authors tried any of them?

-To increase the resolution of the Hsp90-Cdc37-BRAF core, the authors have combined the particles of the three classes (533,127 particles in total). It is hard to think that PP5 binding does not induce any conformational changes at all in the complex, and even some differences would be expected for the two conformational states of PP5. In my opinion, a better resolution could be achieved by performing several rounds of classification in each class, with and without particle alignment, rather than merging all of them together.

REVIEWER COMMENTS

Reviewer #1 (Remarks to the Author):

In this manuscript, Oberoi et al. solved the cryo-EM structures of HSP90-Cdc37-BRAFV600E complex, in the absence or presence of PP5, and demonstrated the ability of PP5 in dephosphorylating Ser/Thr residues in HSP90 and BRAFV600E in the complex. The study provided important molecular insights into the role of PP5 in resetting HSP90 and its kinase clients in the HSP90 chaperone cycle. While only the C-terminal lobe of BRAFV600E is structured in the complex, there is so far only another structure for the HSP90-Cdc37-protein kinase (CDK4) ternary complex. Hence, the study also advances the HSP90 field by offering structural insights into the interaction between HSP90-Cdc37 and kinase clients. Overall, this is a significant study, and the technical quality of the work is excellent. I recommend the acceptance of the manuscript for publication in Nat. Commun. after some revisions:

1. The authors provided phosphoproteomic data to support the capability of PP5 in removing phosphate groups from 2 sites in HSP90, and 12 sites in BRAFV600E in the purified HSP90-Cdc37-BRAFV600E complex in vitro. This part of the study can be strengthened if cellular data supporting this role of PP5 are furnished. For instance, the authors could examine how genetic depletion and overexpression of PP5 in melanoma cells carrying the BRAFV600E mutation modulate the levels of phosphorylation at aforementioned sites in HSP90 and BRAFV600E.

At first sight this seems a very straightforward and sensible request. However, our 'factory reset' model for PP5 function is that it specifically dephosphorylates BRAF^{V600E} that is bound to HSP90-CDC37 prior to release back into the cellular pool, where it can become re-phosphorylated as the cell requires. Consequently, unless the HSP90-CDC37-bound fraction is a substantial part of the total BRAF^{V600E} in the cell and the rate of re-phosphorylation is slow, the change in phosphorylation state is transient and it is going to be difficult to see meaningful differences in the steady-state phosphorylation of the BRAF^{V600E} when PP5 is knocked down or over-expressed.

Nevertheless, we have attempted to look at the effect of depletion and overexpression of PP5 as the referee suggests. Rather than a melanoma cell line, we have used HT29 – a colorectal tumour cell line for which we have a validated stock in house, which harbours a V600E mutant BRAF which we know from multiple other studies, is strongly dependent on HSP90 function. Mass spectrometry analysis of individual phosphorylation

sites with native proteins is not practicable and so we have attempted to do this by western blot using commercially available phosphospecific antibodies. Unfortunately, there are no available phosphospecific antibodies for the HSP90 sites we identified and so we have concentrated on BRAF^{V600E}. We were able to source a phospho-specific antibody from Abcam directed against BRAF-pSer729 – the primary 14-3-3 interaction site, and the site we believe is engaged in the structure of the complex we trapped with the catalytically dead PP5. Although this antibody gave a clean result with the purified HSP90-CDC37-BRAF^{V600E} complex (see the new Figure 5 in the revised m/s) it displayed some non-specific bands in western blots against cell lysates. We were able to partly knock-down, but definitely not knock-out, PP5 in these cells using siRNA, but as predicted were unable to see significant variations in the specific BRAF-pSer729 band in westerns blots from whole cell lysates.

To try and focus in on the fraction of BRAF^{V600E} that is bound to HSP90-CDC37, we tried immunoprecipitating HSP90 from HT29 cells and then western blotted the immunoprecipitates. We were able to see differences in the levels of PP5 that co-IP'd with HSP90 from HT29 cells treated with PP5-siRNA, or overexpressing PP5, and these seemed to correlate with the differences in the amount of BRAF that was detectable in the same HSP90-immunoprecipitates, consistent with a role for PP5 in chaperone regulation of BRAF. However, when we attempt western blots of the immunoprecipitates using the phospho-specific antibody to BRAF-pSer729, any possible signal was completely overwhelmed by non-specificity of the antibody which gives equally strong staining in IP-control and anti-HSP90-IP samples for both PP5-siRNA and PP5 over-expression cells. This non-specificity is depressingly reproducible and has resisted multiple tweaks to the methodology. Unfortunately, to the best of our knowledge this is the only 'phosphospecific' antibody to BRAF-pSer729 that is commercially available.

The results of this considerable amount of work, which has occupied a member of my lab more or less constantly for three months, do not provide the definitive demonstration that the dephosphorylation of BRAF that we can readily demonstrate in the *in vitro* reconstructed system, occurs in cells. However, it is clearly the case from much previously published work that both PP5 and BRAF^{V600E} associate with HSP90 in cells, and from our structural and biochemical data it is also clear that PP5 and BRAF can bind to the same HSP90, and when they do so the PP5 can dephosphorylate the BRAF^{V600E}. So, while we would agree with the referee that a cellular demonstration would strengthen the study, and we would very much have liked to be able to deliver that, we would assert that there is no reason to think that what we see in the *in vitro* reconstruction does not happen in

cells; it is just not possible to visualize it with the reagents currently available to us.

We are reassured that the reviewer thinks our submitted manuscript is a significant study of excellent technical quality, and hope they will not consider the absence of the cellular demonstration a barrier to publication.

I have attached examples of the cellular results we have obtained below, for the reviewer and editor's benefit, but we do not believe they would add anything by being included in the final manuscript.

In addition, the authors should provide, in Supplementary Figures, the annotated MS/MS in supporting the identification of phosphosites in HSP90 and BRAFV600E.

These are huge datasets and we have therefore followed the standard practice in the MS field by depositing them in the PRIDE database, where they can be accessed directly using links provided with the published manuscript. The reviewer can access these datasets as follows :

HSP90-CDC37-BRAF complex - Project accession: PXD033678

Username: reviewer_pxd033678@ebi.ac.uk Password: J7cvvcoP

HSP90-CDC37-CRAF complex - Project accession: PXD035934

Username: reviewer_pxd035934@ebi.ac.uk **Password:** uE0BbLVW

2. In the phosphoproteomic data presented in Supplementary Table, the authors observed elevated phosphorylation of several sites in PP5 after PP5 addition, where appreciable signals for phosphopeptides were also detected for PP5 in the HCB sample without the addition of PP5. Does this suggest that the HCB complex carries PP5 contamination? Since the proteins were purified from insect cells, these results were somewhat surprising.

No, this does not suggest contamination. Although the multiplexed method we used (TMT isobaric tagging) is very reproducible, it has the so called “co-isolation interference” problem that doesn’t permit detection of OFF-ON differences and the dynamic range of quantification is compressed. In other words, there is always some signal even for negative samples depending on the degree of co-isolation. This issue can be avoided with MS3 quantification (<https://doi.org/10.1038/nmeth.1714>), however there is a significant drop in sensitivity in this method especially for low abundant phosphopeptides, therefore we opted for the standard MS2 analysis to profile relative differences.

3. Multiple bands were detected for the HSP90-Cdc37-BRAFV600E–PP5 complex after BS3 crosslinking (Supplementary Figure 1). Can the authors comment on the heterogeneity of the complex?

Unlike some multiprotein complexes that are substantially constitutive, and the reviewer may be more familiar with, the core HSP90-CDC37-kinase and the larger complex with PP5 are highly dynamic transient complexes which are undergoing equilibrium exchange between assembled and disassembled states. The experimental conditions we use seek to stabilise the assembled states we are interested in, but inevitably there is a heterogenous mixture present, which is ‘frozen’ by the addition of crosslinking reagents. The imaging of the single particles that result clearly reflects this heterogeneity, but this turns out to be very helpful, as it has provided three well resolved states that deliver far more biological insight than a single state would have done.

Reviewer #2 (Remarks to the Author):

This manuscript describes cryo-EM structures of HSP90 in complex with the co-chaperone CDC37, protein kinase BRAF V600E and phosphatase PP5 with PP5 in two conformations, and a structure of the HSP90/CDC37/

BRAF V600E complex. In addition, the authors carried out by mass spectrometry experiments to map the potential sites on the different components of the complex that could be de-phosphorylated by PP5. The structures presented here could potentially help to elaborate the mechanism of HSP90, CDC37 or BRAF dephosphorylation by PP5, however the results and proposed mechanism are not supported by any in vitro biological assays, or directly evidenced in the cryo-EM structures – in general, the title and conclusions reached in the manuscript tend to overstate what can be inferred from the data presented. The manuscript would also benefit from extensive editing to improve readability and to improve the clarity of the figures. Overall, the results and structures presented in this manuscript are of general interest to those in the kinase/Ras/RAF signaling field, and may be suitable for publication in Nature Communications, once the following major concerns have been addressed.

Major comments:

- The manuscript tends to overstate the results, or the description of the results is not consistent with or fully supported by the figures. To fully support the statement made by the title ('forms a structural platform for kinase dephosphorylation') and the model presented in Figure 4, additional evidence is needed – either biochemical evidence or further structural evidence.

It is not clear what the reviewer believes falls short of justifying the title, unless it is that we have only shown this for one kinase – BRAF – but have generalised the conclusion to claim that other HSP90-dependent kinases are also processed in a similar way. In that case the reviewer will be happy to relinquish this concern, as in the revised manuscript we now show that CRAF is also substantially dephosphorylated by PP5 in a similar manner. This data is included in the revised manuscript – diagrammatically in Figure 4 and is discussed in the text on page 10 lines 13-19.

Furthermore, as two of the sites dephosphorylated by PP5 in BRAF and now also CRAF are those responsible for binding 14-3-3 protein in important inhibitory complexes, we have looked at the ability of the HSP90-CDC37-BRAF and also CRAF complex to interact with 14-3-3 before and after addition of PP5 to the complex and find that PP5 treatment substantially diminishes the interaction with 14-3-3, suggesting a direct regulatory consequence of PP5 activity on the complex. This data is included in the revised manuscript as Figure 5 and is discussed in the text on page 10 from line 20 onto the top of page 11.

The data presented provides a snapshot of the likely catalytic cycle and does not fully elucidate the dephosphorylation mechanism of PP5.

The catalytic mechanism of PP5 has been extensively described elsewhere – see for example Oberoi et al, PNAS 113, 9009-9014, 2016, as has its requirement for docking to HSP90 in order to access substrates such as CDC37 – see for example Vaughan et al *Mol Cell* **31**, 886-95 2008.

Alternatively, the title and abstract can be changed to better reflect the data presented in the current manuscript. The current structures only show the inactive PP5 binds to the system without engaging the phosphorylated sites in the components.

We do not understand this comment – we make a strong structural case within the resolution limits of this highly dynamic transient structure that the catalytic domain of PP5 – into which we introduced a catalytic mutation previously shown to promote substrate retention – is most likely engaged with pSer729 in the ‘open’ complex. This is presented clearly in FIGURE 3 and in the figure legend thereof, and discussed in the text on page 8 lines 12-19.

- The V600E of BRAF mutant was used throughout the paper. It is briefly discussed that this mutant was used as it appears to be highly HSP90 dependent. Please elaborate, and please discuss how the structure may differ with wild-type BRAF in terms of binding to HSP90.

Again, we do not understand this comment, as we do discuss the likely differences between the interaction of the V600E (and other hydrophilic V600 mutants found in tumours) and the wild type with CDC37 – which confers the specificity for kinases – in terms of the flexibility of the activation segment downstream of the DFG motif which we show interacts with, and is conformationally switched by, the combined action of Arg30 and Trp31. This is clearly illustrated in FIGURE 1E and discussed in detail on page 6 from line 18 onwards.

- The fact that the complex likely is dynamic was referred to repeatedly in the text. Have the authors considered using multi-body refinement (in Relion 4) or 3D variability analysis (in Cryosparc) or 3D classification to better characterize the dynamic states of the complex?

We have of course used a number of different processing strategies in RELION and Cryosparc to try and improve the resolution of the flexible components of the complex, but did not see any substantial improvement – rather there was some loss of resolution. We believe this is most likely due

to the low molecular mass of the most flexible components – the PP5 TPR and phosphatase domain – which certainly makes them unsuited to approaches such as multi-body alignment.

- Interfaces are described for HSP90/CDC37, CCD37/BRAF, but not PP5. Please describe any interfaces between PP5 and other proteins in the complex,

Again, we do not understand this comment, as we clearly discuss the interactions with HSP90 made by the TPR domain of PP5 – illustrated in FIGURE 2C,D and FIGURE 3B and discussed in the legends thereof and in the text on page 7 from line 22 onwards and essentially the whole of page 8.

and also describe the PP5 interactions in the context of the mass spectrometry data.

The ability of PP5 to dock to either of the two C-terminal sites on the HSP90 dimer is fully consistent with the observation that it can dephosphorylate sites in BRAF (and CRAF) that are in the flexible and partly unstructured region N-terminal of the kinase domain that is not ordered in the complex with HSP90, as well as sites in the C-lobe of the kinase and beyond. We show this schematically in FIGURE 6c and discuss this in the figure legend thereof and in the main text on page 13. The two dephosphorylated sites identified within the structurally resolved C-lobe of BRAF are fully surface accessible to the phosphatase domain of PP5 which has substantial flexibility with respect to the rest of the complex through the unstructured linker that connects it to the TPR domain and would have no difficulty in accessing these sites based on the structural observations – this is discussed on page 12, lines 13-17. Finally, the phosphorylation sites just beyond the C-lobe are also well positioned to be accessible to the phosphatase domain. In particular the main 14-3-3 binding site at pSer729 is perfectly positioned to be the site that the catalytically inactive PP5 used for the structural work has been trapped engaged with. In principle the catalytically inactive PP5 phosphatase domain could have been trapped engaged with any of the phosphorylation sites in the BRAF, and it is not impossible that there are small sets particles within the total cryoEM data in which this occurs, but are too rare to be picked up in the cryoEM classification process. The observed position for the phosphatase domain we see predominantly and attribute to engagement with pSer729, is perhaps favoured over other possibilities as it is bolstered by the additional interactions this domain makes with HSP90 when bound in this position with respect to BRAF. This is discussed on page 8 lines 12-19.

- Please include legends for all Supplementary Tables and Supplementary Figures. Please remove all erroneous text on the Supplementary Figures and please include it in the figure legends.

Legends are provided embedded with the Supplementary Figures which we believe is the preferred format for this journal 'family'

- The manuscript will benefit from an initial figure which presents the overall EM maps and structures alongside each other to better orient the reader.

This is provided in Supplementary Figure 2 which we believe is a good place for it.

- Mass spectrometry data is provided that indicate potential dephosphorylation sites for HSP90 and BRAF by PP5. Results are only shown for BRAF in the supplementary figure, and no results are shown for HSP90.

The BRAF sites were presented in schematic form in FIGURE 4A, but in a detailed table in the Supplementary. In the revised manuscript we have expanded the table to also include the sites observed in HSP90, and have included an additional table for the sites we have now found in CRAF. We have now moved these into the main figures.

Please include the results for HSP90 as a figure, and discuss the results closely considering the structural data – are the potential phosphorylation sites phosphorylated in the structures?

The major phosphorylation sites we observe in HSP90 in the HSP90-CDC37-BRAF complex in this study map to the highly disordered linker segment connecting the N-terminal domain to the rest of the protein, and are therefore not resolved.

Given that the mass spectrometry data represents a significant amount of work, please consider including them as a main figure.

We agree that the MS is a major component of the paper, and with the additional MS data on the HSP90-CDC37-CRAF complex we have added, it makes good sense to follow this suggestion, as we have now done in the revised manuscript

- The writing in the manuscript can be somewhat unclear and would benefit from extensive editing to ensure readability and clarity. Please see below

minor comments for detailed suggestions. More thorough and more careful labelling of the figures will aid the reader in interpreting the results.

Minor comments:

- A preprint has been released describing the structure of a Raf-1(CRAF)/HSP90/CDK37 complex (doi: 10.1101/2022.05.04.490607). The structures presented should be compared with the structures presented in this manuscript briefly in the discussion.

We have ourselves determined a cryoEM structure for CRAF, for which a manuscript is in preparation – that will be a more appropriate place for that comparison.

- There is a lot of white space in the figures – please consider arranging the figures to reduce the white space.

I am sure the journal will arrange the panels as they see fit in the published version

- In the manuscript, the terms ‘closed’ and ‘open’ are used interchangeably, referring to both the PP5 and HSP90. This is confusing for the reader, and it would be clearer if different terms are used for the two conformational states of the HSP90/CDC37/BRAF/PP5 complex.

We have attempted to remove any such ambiguities in the revised m/s

- Likewise, when terms like ‘active’ or ‘autoinhibited’ are used, it should be explicit as to if the term is referring to the whole HSP90 complex, or a specific component of the complex.

As above.

- In Supplementary Table 1, please include a map/model FSC value, and include real-space map fitting statistics (i.e. CCmask, CCvolume).

Done in revision

In Supplementary Figure 2, please be explicit about which ab initio models were used to generate the high-resolution models. Please also note that ‘CTF’ should be in all-caps. Please add scale bars for the micrograph and 2D classes in Supplementary Figure 2.

Done in revision

- In Supplementary Figure 3, please re-plot the FSC curves in a suitable plotting program so they are clearer to the reader. Please also include a model-map FSC curve and indicate the FSC=0.5 and 0.143 thresholds in the plot.

Model-map FSC curves have been added to the revised Supplementary Table

- Figures are labelled interchangeably as 'Figure' and 'Fig'. Please be consistent.

Done in revision

- Page 6, line 1 – it is unclear which part of the complex Ser 13 belongs to.
- Page 10, line 18-19 – please note typo 'While the remainder of the N-lobe of CDK4 was partially visible in some subsets of...'.
'

Done in revision

Reviewer #3 (Remarks to the Author):

The manuscript presented by Oberoi et al shows an interesting structural characterization of the mechanism that allows dephosphorylation of BRAF kinase by PP5 with the necessary contribution of Hsp90 and its cochaperone Cdc37. The authors provide three snapshots of the Hsp90-Cdc37-BRAF complex: with PP5 in an autoinhibited and active conformations, and without PP5. The structural results are complemented by functional assays in which the activity of PP5 is assessed by an exhaustive proteomic analysis of the phosphorylation levels of the proteins that form the complex. The significant dephosphorylation observed confirms the activation of PP5 by binding to Hsp90-Cdc37 complex, which results in the inactivation of the oncogenic kinase BRAF.

The results presented here are solidly supported and consistent with previous evidence. The manuscript is well discussed and easy to follow. However, there are two main concerns for me. One of them is that some regions in the CryoEM structures display a limited resolution, which doesn't allow a proper modelling of some key elements like the Hsp90 C-terminal MEEVD. The second one is that the validation reports show a poor fitting of a large portion of Cdc37 and PP5 in both conformations, indicating a lack of density to accommodate these proteins. The authors tackled these issues by docking previously described atomic structures, which is a nice approach to support their proposed mechanism, but can lead to an overinterpretation of the data.

We agree that it is not desirable to make precise comments regarding specific side chain interactions in regions of maps where *ab initio* modelling of side chains is not possible, and we have studiously avoided that in this manuscript, making only 'regional' attributions. In these structures, as in many others, there are substantial variations in the resolution in different regions. Even with sophisticated 'sharpening' techniques such as those used in DeepEMhancer, the resulting map is therefore a compromise. The PDB validation which is based on methodologies developed for X-ray crystallography where local resolution variation is very much smaller, don't really deal with this well.

I would suggest to clarify that these are tentative models that cannot be fully modelled based on the CryoEM data, and remove these regions from the deposited structures.

A very similar situation pertains in the other deposited HSP90-CDC37 kinase complex – that of Verba et al – where the density in many places was also not able to support *ab initio* interpretation. It would be a strange

situation if those entries were to remain with the atomic interpretations, whereas this overall higher resolution study were not to include well fitted prior models in those regions. I believe that the structural biology community, like this reviewer has a very mature understanding of the regional limitations of accuracy in cryoEM maps into which pre-existing experimental structures or indeed alphafold predictions have been docked and refined, and will not be led to draw erroneous conclusions. The vast majority of deposited cryoEM structures fall into this category.

Alternatively, some other experiments could reinforce the proposed models. For instance, XL-MS could confirm interacting regions in medium-resolution regions.

XL-MS – which we have utilised in other studies from our lab – is an inherently low resolution approach that is of greatest value where there is ambiguity of identification of domains or sub-domains within a larger complex. No such ambiguity pertains here where the identity and orientation of the all the domains is unambiguously defined by the density, albeit not at side chain resolution in some places.

There are some other issues that should be addressed:

-The title suggests that the mechanism described is shared among a variety of kinases; however, there is no evidence that supports this. In my opinion, it should be rephrased to a more precise title that describes the particular complex studied here.

We have now demonstrated that CRAF within an HSP90-CDC7 complex is also substantially dephosphorylated by PP5 – the inclusion of this in the revised manuscript should assuage the reviewer's concern.

-Introduction: When BRAFV600E mutant is introduced, it is stated to be highly dependent on Hsp90. A reference should be provided for this.

This has been done

-Supplementary figure 1B shows a biochemical analysis of the complex among the four proteins, with a well-defined peak. It is surprising that in the 3D classification a large proportion of the particles (around 63%) are not bound to PP5, especially when the complex was crosslinked with BS3. This is highly unexpected, and the authors should provide a suitable explanation for this.

Unlike some multiprotein complexes that are effectively constitutive, and the reviewer may be more familiar with, the core HSP90-CDC37-kinase and the larger complex with PP5 are highly dynamic transient complexes which are undergoing equilibrium exchange between open and closed conformations, and between assembled and disassembled states. The experimental conditions we use seek to stabilise the assembled states we are interested in, but inevitably there is a heterogeneous mixture present, which is 'frozen' by the addition of crosslinking reagents. The imaging of the single particles that result clearly reflects this heterogeneity, but this turns out to be very helpful, as it has provided three well resolved states that deliver far more biological insight than a single state would have done.

-Another surprising aspect of the purification of the complex is that two conformations of PP5 are observed, but only one conformation (ATP-closed) is found for Hsp90. Why can't other conformations be found in the sample? Is this due to the presence of molybdate or is it related with the complex formation?

This is most likely due to the molybdate which traps the HSP90 (probably) through formation of a post-ATP hydrolysis ADP-molybdate complex which retains the kinase trapped in the partly unfolded state originally seen in Verba et al and again here. In the absence of molybdate the conformational heterogeneity then extends to the HSP90, and the resultant zoo of conformations – which includes the all-important initial loading state – has proved so far refractory to high-resolution single particle analysis. Believe me - we've tried !!

-General comments for all the figures: I would recommend showing equivalent views of all the complexes, including at least two orthogonal views to visualize all the features of the maps, and indicating the rotation. An image showing the atomic models docked into the final maps would be desirable to assess the quality of the models.

All the maps and data have been deposited in publicly accessible databases from which the interested reader can download them and examine them in a fully interactive 3D format to their heart's content. We do show superimposed maps and fitted atomic models with side chains in Figure 1, and fitted models with secondary structure elsewhere – eg Figure 2B,C,D and Figure 3C.

-Figure 1B: The nucleotide would be better visualized if it were depicted in a different colour. Figure 1C: Ser13 should be highlighted somehow.

Done in revision

-Page 6, lines 1-3: When speaking about Figure 1C: “Ser13... is clearly phosphorylated within the complex and engaged □with the side chains of CDC37 residues His33 and Arg36, and Lys406 of HSP90”. This phosphorylation and its interactions with adjacent residues were already structurally described in Verba et al. (2016), and this should be clearly stated in the text.

Done in revision

-Figure 2B and C: The authors acknowledge that the resolution is insufficient to directly model the C-terminal MEEVD sequence of Hsp90. However, they built an atomic model based on a previous structure. The experimental data doesn't provide any evidence that this region should be modelled as it has been. The text should clearly indicate that this is a tentative model and these residues should be removed from the deposited structure.

This should be taken into account for Figure 3 too.

Done in revision

-Page 8 (final paragraph): The conformational change on PP5 would be better visualized if supported by a figure. It could be included as part of figure 4 or a modified version of Figure 4C.

Done in revision and a 'morph' movie illustrating this has been included with the Supplementary Material

-When doing the image processing, signal subtraction and focused 3D classification without alignment were used to try to improve PP5 resolution. Other tools such as Relion multi-body refinement or cryoSPARC local refinement protocol are very useful approaches for these tasks. Have the authors tried any of them?

We have of course used a number of different processing strategies in RELION and Cryosparc to try and improve the resolution of the flexible components of the complex, but did not see any substantial improvement – rather there was some loss of resolution. We believe this is most likely due to the low molecular mass of the most flexible components – the PP5 TPR and phosphatase domain – which certainly makes them unsuited to approaches such as RELION multi-body.

-To increase the resolution of the Hsp90-Cdc37-BRAF core, the authors

have combined the particles of the three classes (533,127 particles in total). It is hard to think that PP5 binding does not induce any conformational changes at all in the complex, and even some differences would be expected for the two conformational states of PP5. In my opinion, a better resolution could be achieved by performing several rounds of classification in each class, with and without particle alignment, rather than merging all of them together.

We tried a number of different strategies including the conventional approach the reviewer suggested, but this did not produce the same quality as the combine-refine-split pipeline we used – suggested by Basil Greber – which ensured very accurate orientation of the particles' common cores and allowed proper CTF and beam parameter refinement, and effective particle 'polishing'. Subsequent to this the particles were re-classified into the three classes and these were then separately re-classified and re-refined. Thus, the conformational variability of the common regions of the three structures was allowed to fully manifest in the final separate structures.

REVIEWERS' COMMENTS

Reviewer #1 (Remarks to the Author):

The authors have addressed satisfactorily the comments raised by this reviewer from the last review. In particular, the authors made some serious efforts to address whether genetic modulation of PP5 in melanoma cells carrying the BRAFV600E mutation alters the levels of phosphorylation of BRAFV600E. While the outcome of the experiment was not successful, the reviewer understands the limitation of currently available reagents. I recommend the acceptance of the manuscript for publication in Nat. Commun.

Reviewer #2 (Remarks to the Author):

The authors have done well with their rebuttals. I have no further concerns.